# R-GENIE: REASONING-GUIDED GENERATIVE IMAGE EDITING

## ABSTRACT

While recent advances in image editing have enabled impressive synthesis capabilities, current methods remain constrained by explicit textual instructions and simple editing operations, lacking deep comprehension of implicit user intentions and contextual reasoning. In this work, we introduce a novel reasoning-guided generative image editing paradigm, which generates images based on complex, multi-faceted instructions accepting world knowledge and intention inference. To facilitate this paradigm, we first construct a comprehensive dataset featuring 1,070 image-instruction-edit triples that incorporate rich reasoning contexts and world knowledge. We then propose R-Genie: a reasoning-guided generative image editor, which synergizes the generation power of diffusion models with advanced reasoning capabilities of multi-modal large language models. R-Genie leverages a reasoning-attention mechanism to bridge linguistic understanding with visual synthesis, enabling it to handle intricate editing requests involving abstract user intentions and contextual reasoning relations. Extensive experimental results validate that R-Genie can equip diffusion models with advanced reasoning-based editing capabilities, unlocking new potentials for intelligent image synthesis.

## 1 INTRODUCTION

Recent breakthroughs in the community of generative image editing have ushered in a transformative paradigm for progressive visual content manipulation, where natural language instructions enable open and granular image synthesis processes (Cao et al., 2023; Fang et al., 2025; Zhang et al., 2023). The emergence of diffusion-based architectures (Croitoru et al., 2023; Po et al., 2024; He et al., 2025a), *e.g.*, stable diffusion (Rombach et al., 2022) and Imagen (Saharia et al., 2022), has significantly elevated the realism and controllability of the synthesized image quality, making AI-driven editing tools increasingly accessible to non-expert domains (Wu et al., 2024a; Xing et al., 2024). These advancements have spurred widespread adoption across diverse communities, ranging from media production to consumer-driven social media applications (Brooks et al., 2023; Kawar et al., 2023; Shen & Tang, 2024; Shen et al., 2025a), enabling accurate manipulations via textual guidance (Shen et al., 2025b; He et al., 2025b; Jin et al., 2024; Zhao et al., 2025). However, a critical limitation persists in current methods: while they adeptly handle explicit simple edits (*e.g.*, *"Change the dog to a cat."*), their performance deteriorates or even corrupts when confronted with implicit user instructions necessitating world knowledge and contextual reasoning (*e.g.*, *"Identify which animal in the image is panting with its tongue protruding and replace it with a cat."*).

The deep comprehension and faithful execution of user intention in AI systems constitute a crucial step toward achieving artificial general intelligence (Goertzel, 2014; Xie et al., 2024). Achievements in multi-modal large language models (MLLMs) have established these models as versatile interpreters of human intention across multi-modal inputs, including texts, images, videos, and speech (Wu et al., 2024d; Zhang et al., 2024b). These methods exhibit near-human performance in intention understanding, contextual reasoning, and instruction grounding, effectively bridging the semantic gap between high-level user directives and actionable task representations (Caffagni et al., 2024; Fu et al., 2024; Lai et al., 2024; Li et al., 2024). Moreover, MLLMs have shown remarkable capabilities in multi-modal understanding (*e.g.*, image captioning (Bucciarelli et al., 2024), visual question answering (Lee et al., 2024)), visual generation (*e.g.*, text-to-image generation (Wu et al., 2024c), text-guided extrapolation (Fu et al., 2023)), and mixed-modality synthesis (*e.g.*, video keyframe generation conditioned on textual descriptions (Wang et al., 2024b; He et al., 2025b)). However,

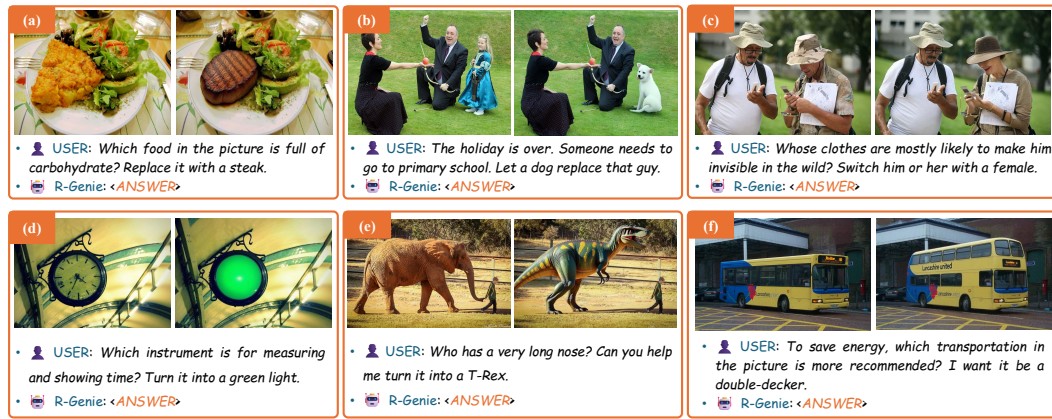

Figure 1: Illustrative samples of our constructed dataset. In contrast to conventional instructions, ours incorporate implicit user intent, offering a more flexible form of guidance.

their deployment in intention-aware and controllable image editing remains in its early stages. More specifically, existing MLLMs face critical challenges in accurately reasoning implicit user intentions (*e.g.*, contextual scene adaptation and pixel-accurate content manipulation) while simultaneously enforcing strict constraints to preserve visual coherence, semantic consistency, and perceptual fidelity in the pixel-accurate synthesized outputs.

These limitations motivate a paradigm shift in generative architecture design to achieve intricate intention comprehension and faithful execution in the image editing task. We propose three fundamental requirements for such an effective solution: (i) sophisticated linguistic parsing of implicit user instructions through MLLMs-based world knowledge (Achiam et al., 2023; Liu et al., 2023; Huang et al., 2024), (ii) strict preservation of visual-semantic constraints during edit operations via contextual reasoning (Lai et al., 2024; Xie et al., 2024; He et al., 2025b), and (iii) adaptive generative refinement via diffusion-based iterative denoising (Huang et al., 2024; Jin et al., 2024; Fang et al., 2025). This solution introduces significant challenges, particularly in simultaneously processing discrete linguistic tokens (*i.e.*, for the textual intention comprehension) and continuous visual representations (*i.e.*, for the visual pixel-level editing) - these modalities traditionally handled by separate networks (Liu et al., 2024; Chen et al., 2025). This integration also presents significant training challenges, as it requires harmonizing discrete token processing for linguistic understanding with continuous-space operations for pixel-accurate visual editing.

In this work, we propose R-Genie, a novel reasoning-guided generative image editing paradigm, which is designed to overcome the limitations of explicit instruction-based editing by incorporating deep reasoning about implicit user intention and contextual visual relations. Specifically, our method builds upon modern diffusion models while integrating advanced reasoning capabilities from MLLMs. Inspired by recent work in knowledge-grounded recognition (Liu et al., 2024) and contextual visual understanding (Xie et al., 2024), we leverage a novel reasoning-attention mechanism that bridges linguistic understanding with visual synthesis. R-Genie inherently encodes complex user intentions through its joint reasoning architecture, eliminating the need for multiple separate refinement networks (Wu et al., 2024b; Chen et al., 2025). To support this task, we also establish a comprehensive dataset containing 1,070 image-instruction-edit triples that incorporate rich world knowledge and reasoning contexts. Consequently, as illustrated in Figure 1 (a), given a simple request with implied intention (*e.g.*, *"Which food in the picture is full of carbohydrate? Replace it with a steak."*), R-Genie performs the visual synthesis via world knowledge. When provided with abstract instructions involving contextual reasoning in Figure 1 (b) (*e.g.*, *"The holiday is over. Someone needs to go to primary school. Let a dog replace that guy."*), R-Genie cal also execute the conceptually appropriate visual edits through its reasoning-guided generation process.

*Quantitatively*, R-Genie achieves superior image editing accuracy compared to existing methods, even when evaluated against models with comparable or larger parameters. *Qualitatively*, R-Genie generates images with more natural pixel-accurate attributes compared to MLLMs while demonstrating a robust understanding of user intention. The obtained results suggest that our method unlocks novel potential for intelligent image synthesis by effectively bridging the gap between high-level user instructions and precise visual realization.

## 2 RELATED WORK

**Diffusion Models for Controllable Image Editing.** Recent years have witnessed remarkable progress in instruction-driven controllable image editing through diffusion models, significantly lowering the barrier to high-quality image manipulation by enabling intuitive natural language control (Zhang et al., 2023; Kawar et al., 2023; Huang et al., 2024). The current landscape can be broadly categorized into two paradigms: *(1)* global semantic-level editing (*e.g.*, style transfer (Karras et al., 2019; Chen et al., 2024) and cross-domain translation (Zhu et al., 2017; Li et al., 2023a)) and *(2)* fine-grained spatial-level editing (*e.g.*, object-level modifications (Brooks et al., 2023) and structural adjustments (Ma et al., 2024)). *Under the first paradigm*, methods such as T2ONet (Shi et al., 2021), Prompt-to-Prompt (Hertz et al., 2022), and Imagic (Kawar et al., 2023) employ latent-space manipulation or attention-based mechanisms to achieve global appearance transformations. While effective for broad stylistic changes, these methods exhibit a critical limitation is that their inability to perform precise, localized modifications, fundamentally restricting their utility in applications requiring pixel-level control (Fang et al., 2025; Gatys et al., 2016; Isola et al., 2017). *Under the second paradigm*, methods such as InstructPix2Pix (Brooks et al., 2023), ReasonPix2Pix (Jin et al., 2024), SmartBrush (Xie et al., 2023), SmartEdit (Huang et al., 2024), ReasonBrain (He et al., 2025b), and MagicBrush (Zhang et al., 2023) integrate instruction conditioning into the diffusion process to achieve spatially precise edits. These methods enable region-specific manipulation while maintaining local consistency (Fu et al., 2023; Hertz et al., 2022). However, their performance remains heavily dependent on the explicitness and accuracy of user-provided instructions, placing a cognitive burden on users to formulate optimal prompts. Despite these advances, existing methods remain constrained by their inability to infer implicit user intent or perform higher-order reasoning (Geng et al., 2024; Chen et al., 2025). Current pipelines lack the capacity to handle multi-faceted queries, incorporate world knowledge, or perform logical inference, which limitations that hinder their robustness in complex real-world editing scenarios (Jin et al., 2024; He et al., 2025b; Jin et al., 2024; Huang et al., 2024). To address these challenges, we propose a new image editing paradigm that integrates MLLMs with diffusion-based editing, endowing the system with advanced reasoning capabilities while preserving fine-grained spatial control. Our method bridges the gap between high-level intention understanding and low-level pixel manipulation, significantly expanding the scope of controllable image editing.

**Multi-Modal Reasoning and Generation Models.** The development of MLLMs and diffusion models has historically followed parallel tracks, where MLLMs excelling in semantic understanding and contextual reasoning, while diffusion models specialized in high-fidelity image synthesis, operating through distinct architectural paradigms (Liu et al., 2023; Zhang et al., 2025; Team, 2024; Xie et al., 2024; Wu et al., 2024d). Fortunately, recent advances have enabled their convergence by aligning conditioning mechanisms (Team, 2024; Zhou et al., 2024), latent feature spaces (Tong et al., 2024; Wang et al., 2024a), and token-level representations (Jiao et al., 2025; Ge et al., 2024), facilitating tighter integration between reasoning and generation. Emerging frameworks such as Chameleon (Team, 2024) and Transfusion (Zhou et al., 2024) bridge reasoning and generation by mapping MLLM outputs to diffusion priors via feature fusion or conditioning alignment. Meanwhile, MetaMorph (Tong et al., 2024), Emu3 (Wang et al., 2024a) and Bagel (Deng et al., 2025) establish a unified token sequence space, enabling dynamic multi-step reasoning alongside fine-grained image refinement. Further innovations like UniToken (Jiao et al., 2025) and SEED-X (Ge et al., 2024) introduce transferable token representations to harmonize diffusion inversion with reasoning outputs. Recently, representative works such as Janus (Wu et al., 2024a) employ bidirectional cross-modal mechanisms to ensure consistency in iterative reasoning and synthesis, while Show-o (Xie et al., 2024) unifies language modeling and the diffusion reverse process within a shared latent space. Despite these architectural advances, current methods remain limited in interpreting complex, context-dependent editing instructions that demand deep semantic reasoning, world knowledge grounding, and multi-step logical inference (Tong et al., 2024; He et al., 2025b; Jin et al., 2024; Huang et al., 2024; Liu et al., 2023). Notably, existing MLLMs exhibit suboptimal performance in pixel-accurate controllable image editing compared to specialized task-specific models, primarily due to insufficient exploration of this domain (Fu et al., 2023; Fang et al., 2025). To address these gaps, we construct a dedicated dataset and introduce a novel understand-then-synthesize paradigm that synergizes MLLMs-driven reasoning with diffusion-based generative refinement, enabling precise, semantically guided image generation. Our method elevates the interpretative capacity of instruction-based image editing while preserving the generative expressiveness of diffusion models, opening new directions for multi-modal understanding and controllable synthesis.

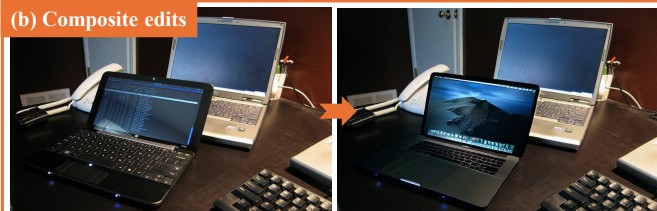

*"Remove the shorter cat in the image who are watching TV."*    *"Among these laptops, mine is running program. I want to replace it with a new model of one."*

Figure 2: Two representative image-instruction-edit triples. **(a))**: atomic edits. **(b))**: composite edits.

## 3 PRELIMINARIES

### 3.1 TASK DEFINITION

The task of reasoning-guided generative image editing involves synthesizing an edited image $X_{\text{edit}}$ given an input image $X_{\text{img}}$ and a high-level textual instruction $X_{\text{txt}}$, which contains implicit reasoning intention. While related to the traditional instruction-based image editing tasks (Fang et al., 2025; Brooks et al., 2023; Geng et al., 2024; Xie et al., 2023), this task introduces three key distinctions: (i) *Complex Query Interpretation*: unlike plain and to the point instructions (*e.g.*, *"Change the male on the right into female."*), as shown in Figure 1 (c), the instructions involve world knowledge (*e.g.*, *"Whose clothes are mostly likely to make him invisible in the wild? Switch him or her with a female."*), *i.e.*, camouflage uniforms are effective for blending into the background outdoors; (ii) *Multi-Step Reasoning*: as shown in Figure 1 (d), the model decomposes abstract intentions (*e.g.*, *"Which instrument is for measuring and showing time? Turn it into a green light."*) into intermediate perception objects (*i.e.*, clocks are used to measure and display time..) and implements semantic edits (*i.e.*, replace the clock with a green light.); and (iii) *Context-Aware Preservation*: as shown in Figure 1, the edited images maintain visual coherence while executing semantically grounded modifications that align with both explicit instructions and inferred contextual constraints.

### 3.2 BENCHMARK DATASET

To facilitate the reasoning-guided generative image editing, we construct REditBench, a comprehensive benchmark dataset for evaluating this task. REditBench consists of 1,070 curated image-instruction-edit triples, addressing the current lack of datasets capable of assessing sophisticated reasoning-based pixel-accurate image editing. The benchmark systematically encompasses two types of edits: **(1) Atomic edits**, as shown in Figure 2 (a), which involves straightforward changes (*e.g.*, *"Remove the shorter cat in the image who are watching TV."*), as in the traditional instruction-based image editing tasks (Xie et al., 2023; Geng et al., 2024; Fang et al., 2025); and **(2) Composite edits**, as shown in Figure 2 (b), which demands reasoning inference and contextual understanding (*e.g.*, *"Among these laptops, mine is running program. I want to replace it with a new model of one."*). The intrinsic distinction between these two types lies in that the latter demands more substantial reasoning capabilities, whereas the former involves directional editing requiring minimal reasoning. These two types of edits simulate real-world editing scenarios where linguistic reasoning and implicit intention are essential for correct execution. The construction of REditBench first utilizes referring image segmentation datasets from RefCOCO/RefCOCO+ (Kazemzadeh et al., 2014) to enable precise spatial localization. High-fidelity edited images are then generated using state-of-the-art inpainting Stable-Diffusion-XL-1.0 (Podell et al., 2023) model, ensuring semantic coherence in the synthesized results. Besides, to mitigate potential biases arising from purely synthetic data, REditBench also incorporates human-annotated edits where professional annotators craft complex editing instructions and corresponding transformation edits following (He et al., 2025b; Lai et al., 2024). This annotation approach guarantees a balanced dataset that reflects both reasoning difficulty and real-world relevance (Zhang et al., 2024a; Jin et al., 2024; Huang et al., 2024). REditBench is partitioned into a *training* set (including 850 samples) and a *val* set (including 220 samples). Rigorous quality assurance is conducted through CLIP-based semantic consistency verification and human review as in (Gannamaneni et al., 2024; Xie et al., 2023; Jin et al., 2024; Otani et al., 2023; Yang et al., 2024; Lai et al., 2024), ensuring that the benchmark reliably assesses a model's ability to interpret and perform reasoning-driven edits. More benchmark details are given in the supplementary material.

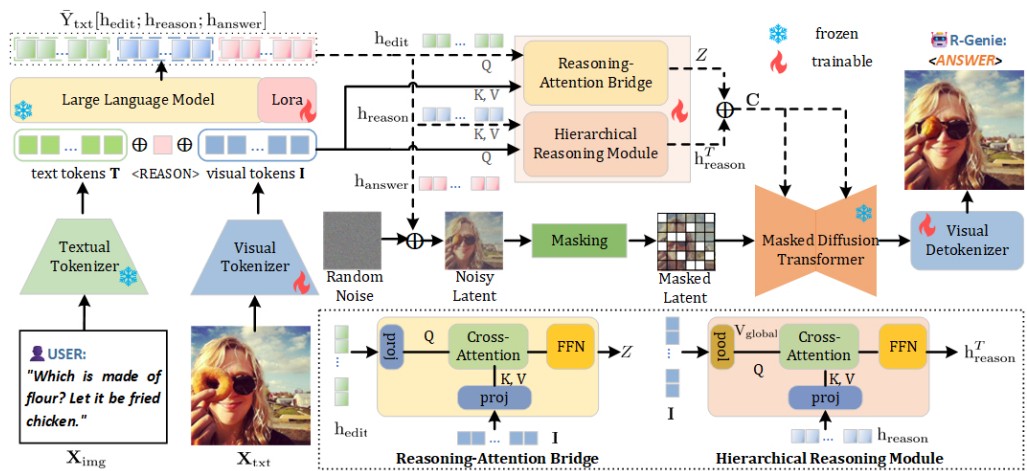

Figure 3: R-Genie introduces an extra `<REASON>` token alongside textual and visual input tokens. The LLM-generated tokens $h_{edit}$ and $h_{reason}$ are subsequently routed through a reasoning-attention bridge and a hierarchical reasoning module, which perform bidirectional reasoning by integrating visual features through cross-modal interactions. The output of these two modules is treated as semantic guidance for diffusion reconstruction. Finally, a masked diffusion transformer accepts $h_{answer}$ and random noise as noisy latent input, and reconstructs the target visual features with mask latent modeling scheme, ensuring alignment between the expected modified visual semantic output and the reconstructed visual representation.

# 4 REASONING-GUIDED GENERATIVE IMAGE EDITOR

As illustrated in Figure 3, R-Genie unifies linguistic reasoning with diffusion-based pixel-accurate image editing model through a novel tokenization scheme and hierarchical architecture. Our method operates in three coordinated stages including tokenization and feature extraction in Sec. 4.1, integrated architecture in Sec. 4.2 and multimodal alignment in Sec. 4.3.

## 4.1 TOKENIZATION AND FEATURE EXTRACTION

Given an input image-instruction pair $(\mathbf{X}_{img}, \mathbf{X}_{txt})$, the visual tokenizer $\mathcal{V}_{enc}$ and the texual tokenizer $\mathcal{T}_{enc}$ first respectively transform the input pair into visual tokens $\mathbf{V}$ and textual tokens $\mathbf{T}$. Here, we introduce a specialized token `<REASON>`, marking the initiation of multi-step reasoning, to govern the reasoning-editing pipeline. The multimodal LLM $\mathcal{G}$ then processes the the obtained tokens through chain-of-thought reasoning, which can be expressed as:

$$\mathbf{I} = \mathcal{V}_{enc}(\mathbf{X}_{img}), \ \mathbf{T} = \mathcal{T}_{enc}(\mathbf{X}_{txt}), \ \hat{\mathbf{Y}}_{txt} = \mathcal{G}(\mathbf{I} \oplus \mathbf{T} \oplus \texttt{<REASON>}), \quad (1)$$

where $\hat{\mathbf{Y}}_{txt} [\mathbf{h}_{edit}; \mathbf{h}_{reason}; \mathbf{h}_{answer}]$ denotes the output tokens of the LLM. $\oplus$ denotes token concatenation. Here $\mathbf{h}_{edit}$, $\mathbf{h}_{reason}$ and $\mathbf{h}_{answer}$ are designed learnable tokens to facilitate subsequent editing and refinement processes. Specifically, $\mathbf{h}_{edit}$ serves as a localized editing signal that encapsulates high-level semantic and spatial instructions for modifying specific regions of the input image. $\mathbf{h}_{reason}$ represents the reasoning pathway embedding, which captures the intermediate rationales generated during the chain-of-thought process. While $\mathbf{h}_{answer}$ functions as the target latent goal for the diffusion process. It embodies the desired latent representation corresponding to the edited outcome.

## 4.2 INTEGRATED ARCHITECTURE

Integrated architecture aims to enable precise and rational instruction-based image editing through hierarchical reasoning and cross-modal attention. To this end, our architecture integrates: a **hierarchical reasoning module** that progressively refines reasoning states through iterative global context integration, a **reasoning-attention bridge** that enables spatially precise cross-modal grounding while reducing over-reliance on global context for explainable and fine-grained edits, and a **masked**

**diffusion transformer** (Peebles & Xie, 2023) under editing constraints from aforesaid modules, enhancing image generation process with the mask latent modeling scheme (Gao et al., 2023).

**Hierarchical Reasoning Module (HRM).** The inputs for HRM are the initial reasoning token $\mathbf{h}_{\text{reason}}^{0}$ and the global image context $\mathbf{I}_{\text{global}}$, producing an output of refined reasoning state $\mathbf{h}_{\text{reason}}^{T}$ (where $T = 1, 2, ..., t, t + 1$ is the dynamic reasoning step length) with the purpose of iteratively updating the reasoning representation through causal inference over visual context. To this end, we perform causal reasoning through stacked transformer layers, which can be expressed as:

$$\mathbf{h}_{\text{reason}}^{t+1} = \text{TransformerBlock}(\mathbf{h}_{\text{reason}}^{t} \| \mathbf{I}_{\text{global}}), \tag{2}$$

where $\mathbf{I}_{\text{global}} = \frac{1}{HW} \sum_{i,j} \mathbf{I}_{ij}$ is the pooled tokens, and $t$ represents the dynamic reasoning step index.

**Reasoning-Attention Bridge (RAB).** With the editing visual token $\mathbf{h}_{\text{edit}}$ and spatial visual features $\mathbf{I}$ as its inputs, HRM generates a spatially-grounded editing feature $\mathbf{Z}$ as output for the purpose of enabling fine-grained, cross-modal modulation of visual features based on inferred editing instructions. This bridge is used to compute spatially-aware editing weights, which can be formulated as:

$$\alpha_{ij} = \text{softmax}\left(\frac{(\mathbf{W}_Q \mathbf{h}_{\text{edit}})(\mathbf{W}_K \mathbf{I}_{ij})^\top}{\sqrt{d}}\right), \quad \mathbf{Z} = \sum_{i,j} \alpha_{ij} \cdot \mathbf{W}_V \mathbf{I}_{ij}, \tag{3}$$

where $\mathbf{W}_*$ denoting projection matrices and $d = 1024$. After fine-grained semantic enhancement within **HRM** and **RAB**, the model achieves improved alignment between high-level reasoning and localized image edits. Then, final reasoning states $\mathbf{h}_{\text{reason}}^{T}$ and the grounded edits $\mathbf{Z}$ will be expressed as $\mathbf{C}$ through a feature-wise concatenation as guidance and constraints for the upcoming diffusion reconstruction stage, which can be expressed as:

$$\mathbf{C} = \mathbf{h}_{\text{reason}}^{T} \oplus \mathbf{Z}, \tag{4}$$

**Diffusion Transformer with Masking Latent Modeling.** To reconstruct the target visual features $\mathbf{h}_{\text{answer}}$, we employ the masked diffusion Transformer as the image synthesizer (Peebles & Xie, 2023). As the multimodal LLM $\mathcal{G}$ employs discrete image tokens, we perform diffusion modeling on the discrete space following (Geng et al., 2024; Fang et al., 2025). For modeling visual tokens $\mathbf{h}_{\text{answer}} = \{v_1, v_2, ..., v_N\}$, we randomly replace the image tokens with the `<MASK>` token in the forward diffusion process. This scheme can force the model to learn the relations among image latent tokens, particularly the associated relations among semantic parts in an image (Zhang et al., 2023; He et al., 2025a). The denoising process reconstruct the original image token using unmasked regions by maximizing the masked token prediction likelihood. This process is formulated as:

$$L_{\text{reconstruct}} = \underbrace{\sum_{i=1}^{N} p_\theta(v_i | \tilde{v}_1, \tilde{v}_2, ..., \tilde{v}_N, \mathbf{C})}_{\text{Next Token Prediction}} + \underbrace{\sum_{j} q_\theta(v_j | \tilde{v}_*, \tilde{v}_2, ..., \tilde{v}_*, ..., \tilde{v}_N, \alpha_\tau)}_{\text{Masked Token Modeling}}, \tag{5}$$

where $\alpha_\tau$ are the diffusion noise schedule parameters, $p_\theta$ and $q_\theta$ model the conditional probability of denoised tokens and masked tokens respectively, $\{\tilde{v}_1, \tilde{v}_2, ..., \tilde{v}_N\}$ represents noised latent tokens and $\tilde{v}_*$ represents masked tokens. The denoise process employs autoregressive transformer layers to predict the masked tokens as in (Xie et al., 2024).

## 4.3 Training

During training, to fully bridge text-image modalities while addressing the potential conflict between contrastive and reconstruction objectives, we develop a hybrid alignment mechanism. The mechanism initiates with Show-o-pre-trained encoders (Xie et al., 2024) for both modalities (visual tokens $\mathbf{I}$ and textual tokens $\mathbf{T}$) as in VILA-U (Wu et al., 2024d). The core challenge for this mechanism arises from the divergent feature requirements, *i.e.*, the contrastive learning ($L_{\text{con}}$) demands high-level semantic alignment, whereas the reconstruction ($L_{\text{reconstruct}}$) relies on low-level visual fidelity (Zhang & Cheng, 2025). To address this challenge, our solution integrates two training tricks. First, we freeze the texual tokenizer $\mathcal{T}_{\text{enc}}$ and allow a trainable vision tokenizer, which can preserve stable semantic anchors with the intention of remaining stable during training. Then, we contrive final loss function with adaptive balancing scheme. The optimal trade-off between semantics and detail evolves as the

model learns; early on, strong contrastive guidance is crucial to establish a robust semantic foundation, while later stages benefit from an increased focus on pixel-level accuracy to refine the output. The scheme offers the benefit of preventing either objective from dominating prematurely, leading to more stable convergence and a superior final model that excels at both semantic understanding and high-fidelity generation. This can be formalized can be formalized as follows:

$$\mathcal{L}_{\text{total}} = \underbrace{\lambda_1 \cdot \text{InfoNCE}(\mathbf{I}_{\text{global}}, \mathbf{T})}_{\text{Contrastive}} + \underbrace{\lambda_2 \cdot L_{\text{reconstruct}}}_{\text{Reconstruction}}, \tag{6}$$

where $\lambda_1 = 1 - \alpha_t$, $\lambda_2 = \alpha_t$ are time-dependent coefficients ($\alpha_t$ follows the diffusion noise schedule), and InfoNCE is the contrastive loss between visual tokens and textual tokens.

## 5 EXPERIMENTS

### 5.1 EXPERIMENTAL SETTING

**Implementation Details.** We adopt the lightweight Show-o (1.3B) proposed in (Xie et al., 2024) as the baseline, where Phi-1.5 (Li et al., 2023b) is used as the core large language model. We use two NVIDIA GeForce RTX 3090 GPUs in training, employing DeepSpeed with ZeRO optimization alongside FP16 mixed precision to significantly reduce memory overhead while maintaining computational efficiency. $\alpha_t$ is set to 0.5 as in (Lai et al., 2024; Xie et al., 2024). The AdamW optimizer (Loshchilov & Hutter, 2017) is used with a learning rate of 3e-4 with weight decay of 0 for parameter updates. For a fair comparison, the training procedure consists of 100 epochs. Unless otherwise specified, all hyperparameters follow the same configuration as LISA (Lai et al., 2024).

**Datasets.** As outlined in Sec. 3.2, the proposed REditBench is methodologically derived from the RefCOCO (Kazemzadeh et al., 2014) benchmark to systematically evaluate the reasoning editing task (Huang et al., 2024). Our annotation process achieved high label consistency (Cohen's Kappa=0.86) while preserving diversity across 70+ scene categories. It is worth mentioning that as we fine-tune LLMs that already possess substantial world knowledge, the limited dataset suffices to specialize the reasoning patterns for editing.

**Evaluation Metrics.** We utilize a systematic evaluation framework that assesses the model's performance. The evaluation is conducted using the following four principal metrics including CLIP Score ($\text{CLIP}_{\text{Score}}$) (Radford et al., 2021), Background L1 & L2 Distance ($\text{L1}_{\text{Bg}}$ & $\text{L2}_{\text{Bg}}$), LAION Aesthetic Predictor score (AP) (Schuhmann et al., 2022), and RISEBench Score (RISE-BS) (Zhao et al., 2025).

### 5.2 REASONING IMAGE EDITING COMPARATIVE ANALYSIS

We conduct a systematic evaluation comparing our R-Genie with nine state-of-the-art methods, categorized into two groups: (1) task-specific instruction-based editing models (*i.e.*, InstructPix2Pix (Brooks et al., 2023), MagicBrush (Zhang et al., 2023), MGIE (Fu et al., 2023), InstructDiffusion (Geng et al., 2024), SmartEdit (Huang et al., 2024)), EditWorld (Yang et al., 2024), X2Edit (Ma et al., 2025), Grounding DINO (Liu et al., 2024)(with stable diffusion models), GLIGEN (Li et al., 2023c), and (2) unified multimodal models (*i.e.*, Show-o (Xie et al., 2024), Janus (Wu et al., 2024a), VILA-U (Wu et al., 2024d), OmniGen (Xiao et al., 2024)), and SEED-X (Ge et al., 2024). All experiments are conducted on our proposed REditBench under identical conditions to ensure fair result comparisons. Quantitative results in Table 1 reveal three key findings: *First*, regarding editing precision, R-Genie achieves state-of-the-art performance in CLIP Score (62.14%) and background preservation ($\text{L1}_{bg}$ Distance: 0.0602, $\text{L2}_{bg}$ Distance: 0.0201), with a 1.8% ,7.2% and 8.2% respective improvement over the second-best methods (*i.e.*, VILA-U and Janus respectively). This demonstrates our method's superior capability in maintaining semantic alignment while minimizing unintended modifications - a critical requirement for reasoning-intensive edits. *Second*, analysis of model efficiency shows that R-Genie attains these results with only 1.3B parameters, outperforming larger models like SmartEdit (7.0B) VILA-U (7.0B), and SEED-X (17.0B) in most metrics. This efficiency comes from our hybrid alignment paradigm that strategically integrates LLM-based semantic parsing with diffusion processes. *Third*, comparative analysis across different architectural approaches reveals fundamental performance trade-offs: conventional instruction-based methods (*e.g.*, InstructPix2Pix (Brooks et al., 2023)) exhibit limitations in handling compositional reasoning due to their reliance on standard text

| Method | #Params | Type | Stage | CLIP$_{Score}$↑ | L1$_{Bg}$↓ | L2$_{Bg}$↓ | AP↑ | RISE-BS↑ |
|---|---|---|---|---|---|---|---|---|
| SEED-X (Ge et al., 2024) | 17.0B | UNI | ONE | 60.79 | 0.0769 | 0.0363 | 4.56 | 62.5 |
| EditWorld (Yang et al., 2024) | 8.0B | SPE | TWO | 48.09 | 0.1005 | 0.0399 | 3.56 | 47.7 |
| MagicBrush (Zhang et al., 2023) | 7.8B | SPE | ONE | 52.22 | 0.1327 | 0.0465 | 3.90 | 51.8 |
| MGIE (Fu et al., 2023) | 7.0B | SPE | ONE | 57.31 | 0.0970 | 0.0364 | 4.39 | 43.5 |
| VILA-U (Wu et al., 2024d) | 7.0B | UNI | ONE | 61.07 | 0.1103 | 0.0533 | 2.74 | 53.3 |
| SmartEdit (Huang et al., 2024) | 7.0B | SPE | ONE | 60.98 | 0.0671 | 0.0245 | **4.67** | 62.7 |
| Grounding DINO-LDM (Liu et al., 2024) | 6.6B | SPE | TWO | 43.06 | 0.1187 | 0.0422 | 4.38 | 50.9 |
| InstructPix2Pix (Brooks et al., 2023) | 4.1B | SPE | ONE | 56.86 | 0.1334 | 0.0476 | 4.51 | 51.7 |
| OmniGen (Xiao et al., 2024) | 3.8B | SPE | ONE | 59.85 | 0.0682 | 0.0237 | 4.08 | 60.9 |
| GLIGEN-LDM (Li et al., 2023c) | 3.7B | SPE | TWO | 45.64 | 0.1045 | 0.0372 | 4.68 | 51.4 |
| InstructDiffusion (Geng et al., 2024) | 2.2B | SPE | ONE | 34.27 | 0.1064 | 0.0425 | 4.35 | 40.6 |
| Show-O (Xie et al., 2024) | 1.3B | UNI | ONE | 47.37 | 0.0855 | 0.0421 | 3.76 | 43.5 |
| Janus (Wu et al., 2024a) | 1.3B | UNI | ONE | 24.30 | 0.0649 | 0.0219 | 2.42 | 41.2 |
| X2Edit (Ma et al., 2025) | 0.9B | SPE | ONE | 42.44 | 0.1091 | 0.0412 | 2.53 | 42.2 |
| **R-Genie(Ours)** | 1.3B | SPE | ONE | **62.14** | **0.0602** | **0.0201** | 4.64 | **64.0** |

Table 1: Quantitative result comparisons with SOTA methods. "SPE" denotes the model is a specific method. "ONE" and "TWO" indicate whether the model is a single-stage or two-stage method.

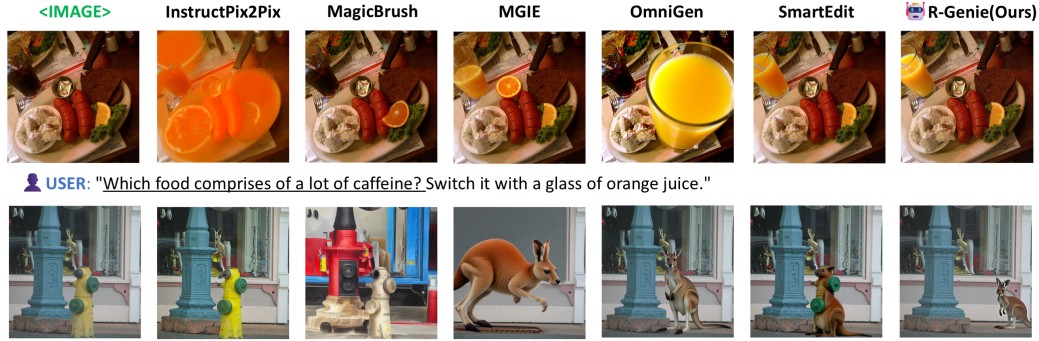

Figure 4: Qualitative result comparisons with SOTA instruction-based image editing methods.

tokenizers; direct MLLM-diffusion coupling approaches (*i.e.*, MGIE (Fu et al., 2023) and Omni-Gen (Xiao et al., 2024)) demonstrate optimization instability, as evidenced by their higher $L2_{BG}$ Distance; while autoregressive architectures (*i.e.*, Janus (Wu et al., 2024a) and VILA-U (Wu et al., 2024d)) present inherent constraints in detail synthesis, reflected in their lower AP scores. This systematic comparison highlights how each architectural paradigm addresses - or fails to address - the critical challenges in reasoning-aware image editing.

## 5.3 QUALITATIVE RESULT COMPARISONS

Figure 4 illustrates that existing approaches frequently misinterpret compositional instructions, leading to two primary failure cases. The first failure case, *target misidentification*, arises when models incorrectly localize objects described by abstract attributes (*e.g.*, *"food comprising a lot of caffeine"*). For instance, when given the instruction *"Which food comprises of a lot of caffeine? Switch it with a glass of orange juice"*, baseline models such as InstructPix2Pix (Brooks et al., 2023), MGIE (Fu et al., 2023), OmniGen (Xiao et al., 2024) erroneously modify non-fire hydrant area, indicating their inability to properly interpret the phrase *"food comprising a lot of caffeine"* as referring specifically to an *"coke."* This limitation extends to other similar cases where models struggle with attribute-based object identification. The second failure case, *instruction-output incongruity*, is featured by implausible artifacts or disrupted object relationships. For example, SmartEdit (Huang et al., 2024) produces unrealistic outputs (*e.g.*, *"a kangaroo with hydrant body"* when instructed to *"replace the hydrant with a kangaroo"* failing to maintain coherent anatomical structures. In contrast, R-Genie grounds the instruction in commonsense knowledge before performing spatially aware edits in the generative process, leading to more accurate and coherent results. More visual comparisons will be concluded in the supplementary material.

Table 2: Results of the ablation analysis and user study. "Baseline" denotes the Show-o (1.3B) (Xie et al., 2024). "Pretrain" and "HA" denote the Show-o-pre-trained encoders and the hybrid alignment mechanism as introduced in Sec. 4.3. HRM & RAB are our proposed hierarchical reasoning module and reasoning-attention bridge in Sec. 4.2.

| (a) Results of the ablation study | | | | | | | (b) Results of the user study | |
|---|---|---|---|---|---|---|---|---|
| Baseline | Pretrain | HA | HRM & RAB | $CLIP_{Score}\uparrow$ | $L2_{Bg}\downarrow$ | $L1_{Bg}\downarrow$ | **Method** | Frequency |
| ✓ | ✗ | ✗ | ✗ | 47.37 | 0.1246 | 0.0421 | InstructPix2Pix | 31/220 |
| ✓ | ✓ | ✗ | ✗ | 53.29 | 0.1207 | 0.0418 | OmniGen | 63/220 |
| ✓ | ✓ | ✗ | ✓ | 42.99 | 0.1335 | 0.0534 | MGIE | 5/220 |
| ✓ | ✓ | ✓ | ✓ | **62.14** | **0.0602** | **0.0201** | **R-Genie (Ours)** | 200/220 |

## 5.4 ABLATION ANALYSIS

To rigorously evaluate our paradigm's architectural contributions, we conduct a systematic ablation study by progressively integrating design components into the Show-o 1.3B baseline model (Xie et al., 2024). The experimental results, detailed in Table 2, demonstrate three key findings. *First*, our procedurally-generated synthetic training data yields significant improvements on CLIP similarity (*i.e.*, +8.7% on reasoning-focused metrics), validating the importance of high-quality training data for instruction-based editing tasks. *Second*, naive visual-textual alignment through direct loss minimization proves unstable (*i.e.*, diverging in 67% of trials), necessitating constrained optimization via the proposed HRM and RAB for stable unimodal feature alignment. *Third*, our hybrid optimizing strategy, which freezes the text encoder while selectively fine-tuning visual encoder layers, achieves an optimal balance, preserving 98.2% of baseline knowledge while improving visual grounding by +12.3%. The complete paradigm establishes new state-of-the-art performance (*i.e.*, +15.2% average improvement), demonstrating the compound benefits of our key innovations: (i) semantically-diverse synthetic data, (ii) regularized cross-modal alignment objectives, and (iii) selective parameter adaptation. These results empirically confirm our core design principle that effective instruction-grounded multimodal learning requires carefully coordinated optimization across data, alignment, and adaptation mechanisms to achieve robust performance.

## 5.5 USER STUDY

To quantitatively assess the effectiveness of our approach, we design a comprehensive user study following rigorous evaluation protocols. Specifically, for each evaluation instance, 22 participants are presented with: (1) a source image, (2) a corresponding reasoning instruction, and (3) three edited outputs from anonymous candidates, including InstructPix2Pix (Brooks et al., 2023), MGIE (Fu et al., 2023), OmniGen (Xiao et al., 2024) and R-Genie. Participants are instructed to select all results they deem satisfactory based on two key criteria: visual fidelity and semantic alignment with the given instruction. This multi-select paradigm enables a more nuanced evaluation of user preferences. As demonstrated in Table 2, our method consistently outperforms competing approaches across both evaluation metrics, achieving statistically significant preference rates. These empirical results strongly validate our method's superior reasoning capacity and generation quality, particularly in maintaining instruction-intent consistency while preserving realistic image characteristics.

## 6 CONCLUSION AND FUTURE WORK

In this work, we introduced R-Genie, a reasoning-guided generative framework for image editing that tackles the fundamental challenges of converting knowledge-rich instructions into semantically coherent visual outputs. R-Genie integrated a reasoning mechanism to bridge the conceptual reasoning capabilities of multimodal large language models with diffusion-based generative control. This synergy enabled fine-grained interpretation of implicit user intent while maintaining context-aware reasoning fidelity. To facilitate this task, we also construct a comprehensive benchmark dataset REditBench. Extensive experiments validated that R-Genie augments diffusion models by infusing structured reasoning into the editing pipeline, thereby achieving robust performance on inherently ambiguous and multi-faceted natural language queries. Building on our current framework, in the future, we plan to investigate the method's effectiveness in more fine-grained settings while extending its application to video-based scenarios, which would advance reasoning-guided generative editing techniques.

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
