# Supplementary Material for paper:
# R-Genie: Reasoning-Guided Generative Image Editing

To enhance the reproducibility and transparency of our work, we present additional dataset samples in Section A1. We also provide comprehensive visualizations of the ablation study results in Section A2, along with a comparative analysis against contemporary unified multimodal understanding and generation approaches in Section A3. After that, more experiments on runtime evaluation are provided in Section A4, meanwhile more convincing qualitative comparison results are illustrated in Section A5. Furthermore, we provide details about user study results in Section A6.

## A1 Dataset Details

As detailed in Section 3.2, our dataset features curated instruction-image-edit triples for image editing tasks, where each sample incorporates natural language instructions requiring compositional reasoning, source and target image pairs. Here we present more triples examples of REditBench in Figure A1. Besides, the comparison with related datasets is presented in the Table A1. It is also worth mentioning that our dataset is released under the CC BY-NC 4.0 (research-only) with commercial use requiring authorization.

**Limitations in domain coverage.** While our approach generalizes well to in-distribution edits (*e.g.*, modifying common objects like "persons" or "dogs"), performance may degrade for highly specialized domains (*e.g.*, medical imaging or rare artistic styles) where training data is scarce. Future work could incorporate domain adaptation methods to mitigate this limitation.

**The potential bias.** Potential biases stem from template structure (prioritizing certain reasoning patterns) and annotator tendencies (*e.g.*, frequent attribute combinations like "red car"). We mitigated this via diverse annotators but acknowledge some cultural/linguistic biases may persist.

Table A1: Comparison of different datasets.

| Dataset | Controllable | Reasoning | Size | Open Access |
|---|:---:|:---:|:---:|:---:|
| InstructPix2Pix (Brooks et al., 2023) | ✓ | ✗ | 454,445 | ✓ |
| Reason50K (He et al., 2025) | ✓ | ✓ | 51,039 | ✗ |
| ReasonPix2Pix (Jin et al., 2024) | ✓ | ✓ | 40,212 | ✗ |
| MagicBrush (Zhang et al., 2023) | ✓ | ✗ | 10,388 | ✓ |
| EditWorld (Yang et al., 2024) | ✓ | ✗ | 10,000+ | ✓ |
| RISEBench (Zhao et al., 2025) | ✓ | ✓ | 360 | ✓ |
| ReasonEdit (Huang et al., 2024) | ✓ | ✓ | 219 | ✓ |
| REditBench (Ours) | ✓ | ✓ | 1,070 | ✓ |

## A2 Visualization of Ablation Study Results

To rigorously evaluate the performance gains attributable to various components of our proposed paradigm, Figure A2 presents a comprehensive ablation study through comparative visual analysis. The systematic integration of individual architectural elements (from left to right) demonstrates statistically significant improvements in the model's cross-modal reasoning capabilities. Quantitative metrics confirm that each progressive enhancement: (1) elevates semantic alignment accuracy between

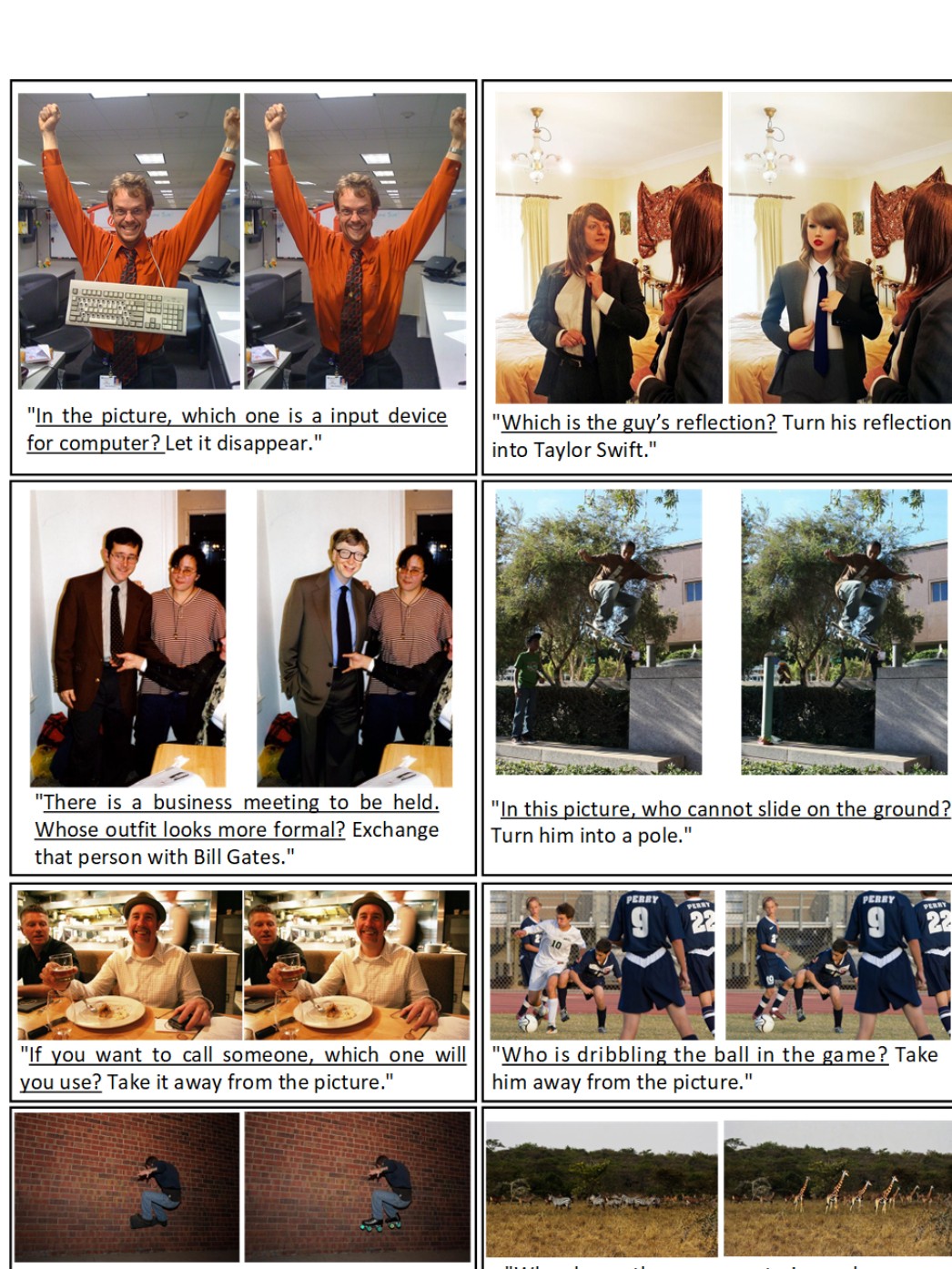

Figure A1: More Examples of the annotated image-instruction-edit triples.

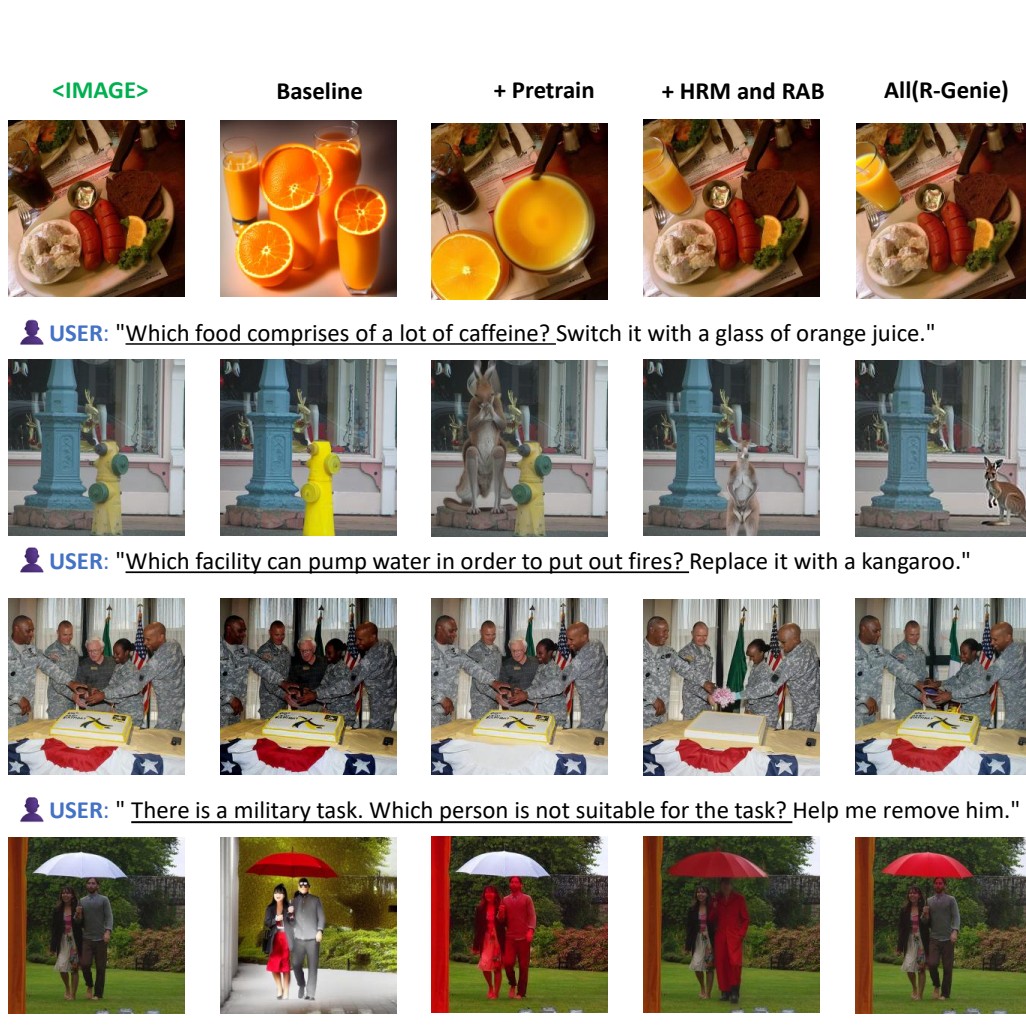

Figure A2: Visualization of Ablation Study Results.

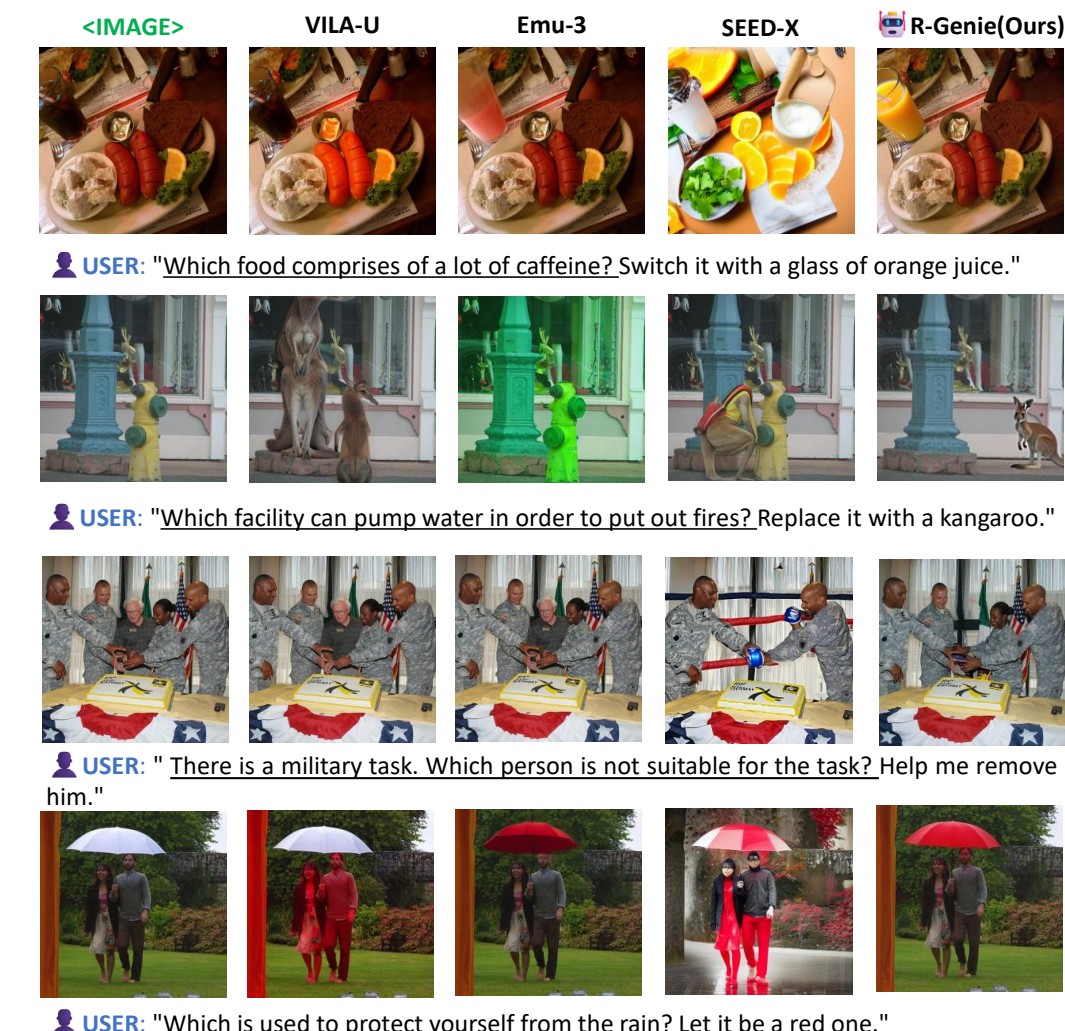

Figure A3: Qualitative comparison with unified multimodal understanding and generation methods.

input modalities, and (2) enhances perceptual coherence in synthetic outputs. These empirical results validate our design choices while providing insights into the relative contributions of each module.

## A3 COMPARATIVE ANALYSIS WITH UNIFIED MULTIMODAL UNDERSTANDING AND GENERATION METHODS

To ensure comprehensive comparative analysis, we extend our evaluation to benchmark general multimodal understanding and generation frameworks (Wu et al., 2024; Xiao et al., 2024; Wang et al., 2024; Ge et al., 2024), despite their inherent limitations in being specifically optimized for instruction-guided image editing tasks. As demonstrated in Figure A3, VILA-U (Wu et al., 2024) exhibits fundamental deficiencies in target object recognition across most test samples, indicating critical limitations in visual grounding capabilities. However, Emu-3 (Wang et al., 2024) produces outputs with marginal modifications relative to source images, revealing constrained multimodal reasoning and adaptive generation capacities. SEED-X has shown its identification capabilites in certain scenarios and is still limited in background perseverance. These comparative observations collectively illustrate the technical challenges in achieving robust integration of semantic understanding and precise image manipulation within current multimodal frameworks.

Table A2: Comparison of different datasets.

| Methods | Runtime (generating one sample) | Param |
|---|---|---|
| InstructPix2Pix (Brooks et al., 2023) | 26.6s | 4.1B |
| OmniGen (Xiao et al., 2024) | 24.6s | 3.8B |
| SmartEdit (Huang et al., 2024) | 43.5s | 7.0B |
| RGenie(Ours) | 14.7s | 1.3B |

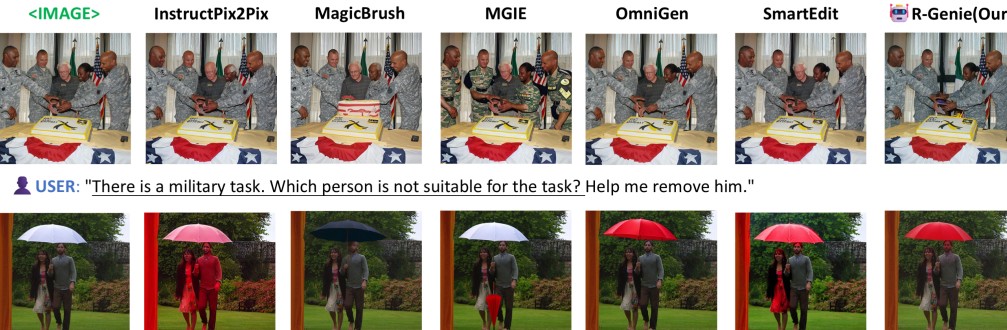

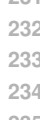
USER: "There is a military task. Which person is not suitable for the task? Help me remove him."

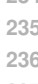
USER: "Which is used to protect yourself from the rain? Let it be a red one."

Figure A4: More visual comparison results.

## A4 MORE EXPERIMENTS

In this section, we compared the runtime required for the model to generate one sample. As shown in R-Table A2, our obtained results (evaluated by two NVIDIA GeForce RTX 3090 GPUs) reveal that R-Genie's runtime is highly competitive: requiring only 14.7s per sample under the use of Phi-1.5 (Li et al., 2023). This is significantly faster than comparable methods like InstructPix2Pix (26.6s), OmniGen (24.6s), and SmartEdit (43.5s), while delivering substantial accuracy gains as demonstrated in Table 1 of the main paper.

## A5 MORE QUALITATIVE COMPARISON RESULTS

To present more convincing and comprehensive qualitative comparisons, here we provide more editing samples in Figure A4.

## A6 USER STUDY RESULTS

To evaluate the efficacy of our methodology, we conducted a comprehensive user study as detailed in the Experiment section. Each response alternative in the survey corresponds to one of the compared methods: Option A denotes outputs from InstructPix2Pix (Brooks et al., 2023), Option B represents results generated by MGIE (Fu et al., 2023), and Option C indicates outcomes from R-Genie. The aggregated results presented in Figures A5 to A10 demonstrate that R-Genie achieved statistically significant preference among participants. This empirical validation substantially supports our method's superiority in human perceptual evaluation compared to existing baseline approaches.

## REFERENCES

Tim Brooks, Aleksander Holynski, and Alexei A Efros. Instructpix2pix: Learning to follow image editing instructions. In *Proceedings of the IEEE/CVF conference on computer vision and pattern recognition*, pp. 18392–18402, 2023. 1, 5

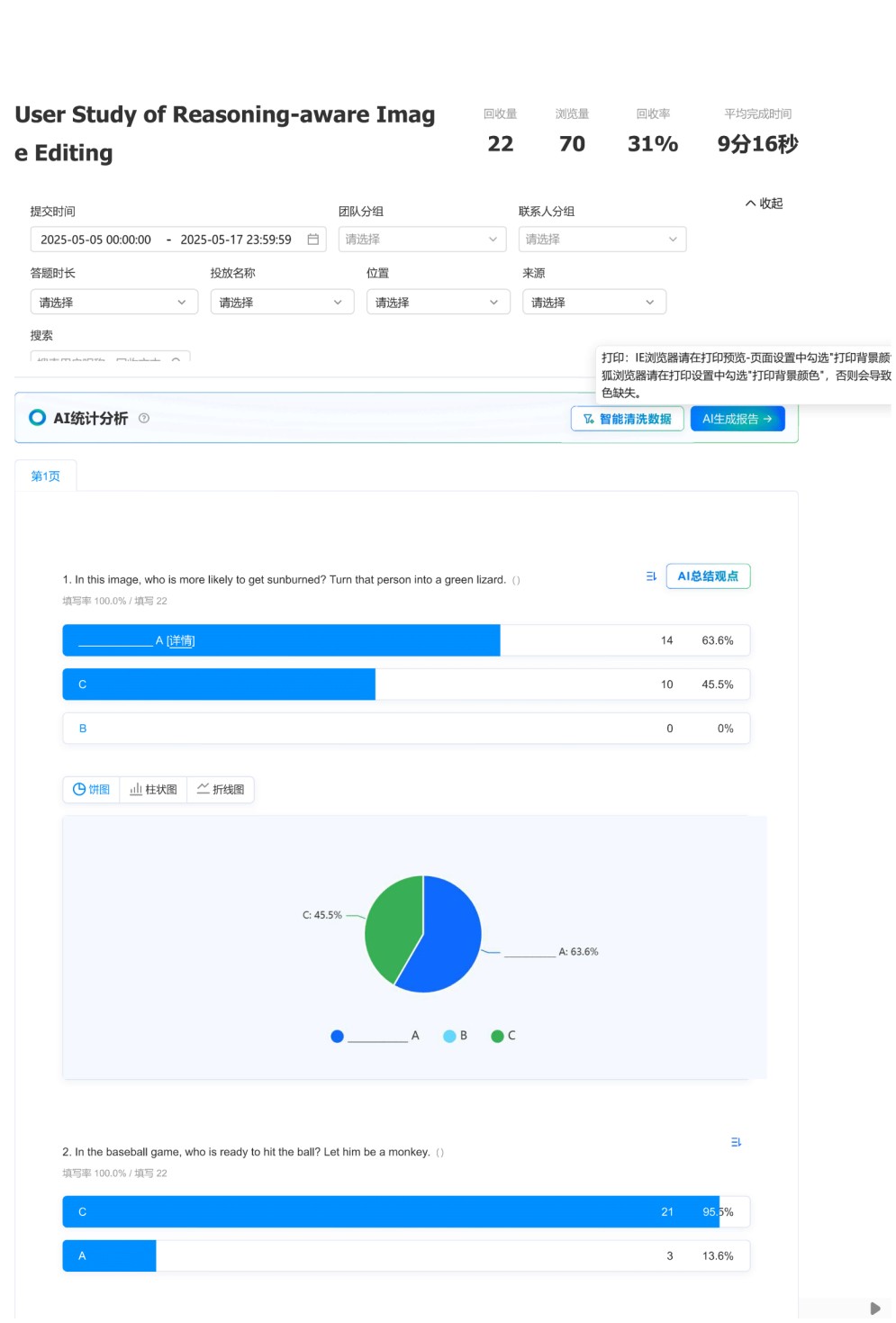

Figure A5: User Study Snapshot: Page 1

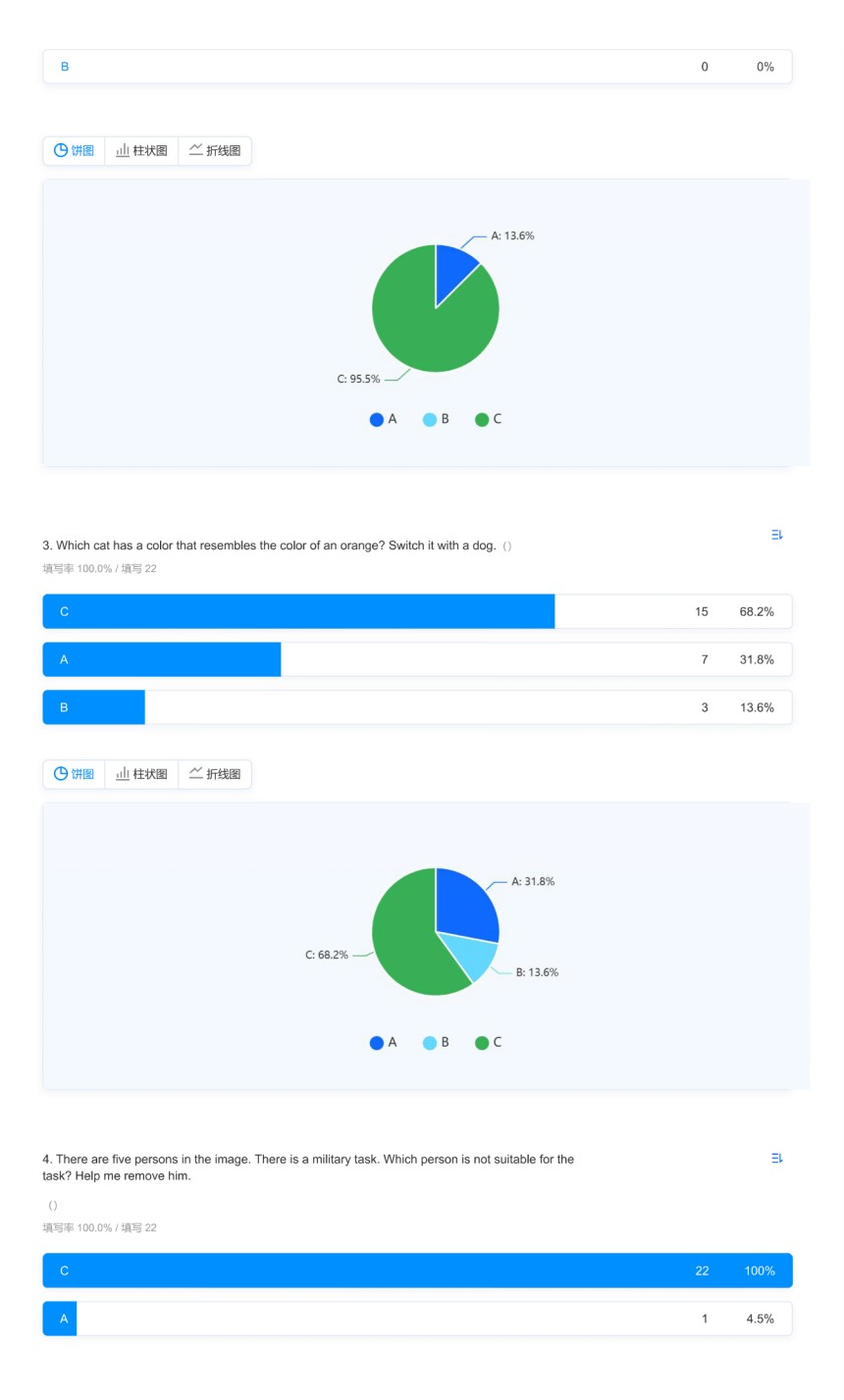

Figure A6: User Study Snapshot: Page 2

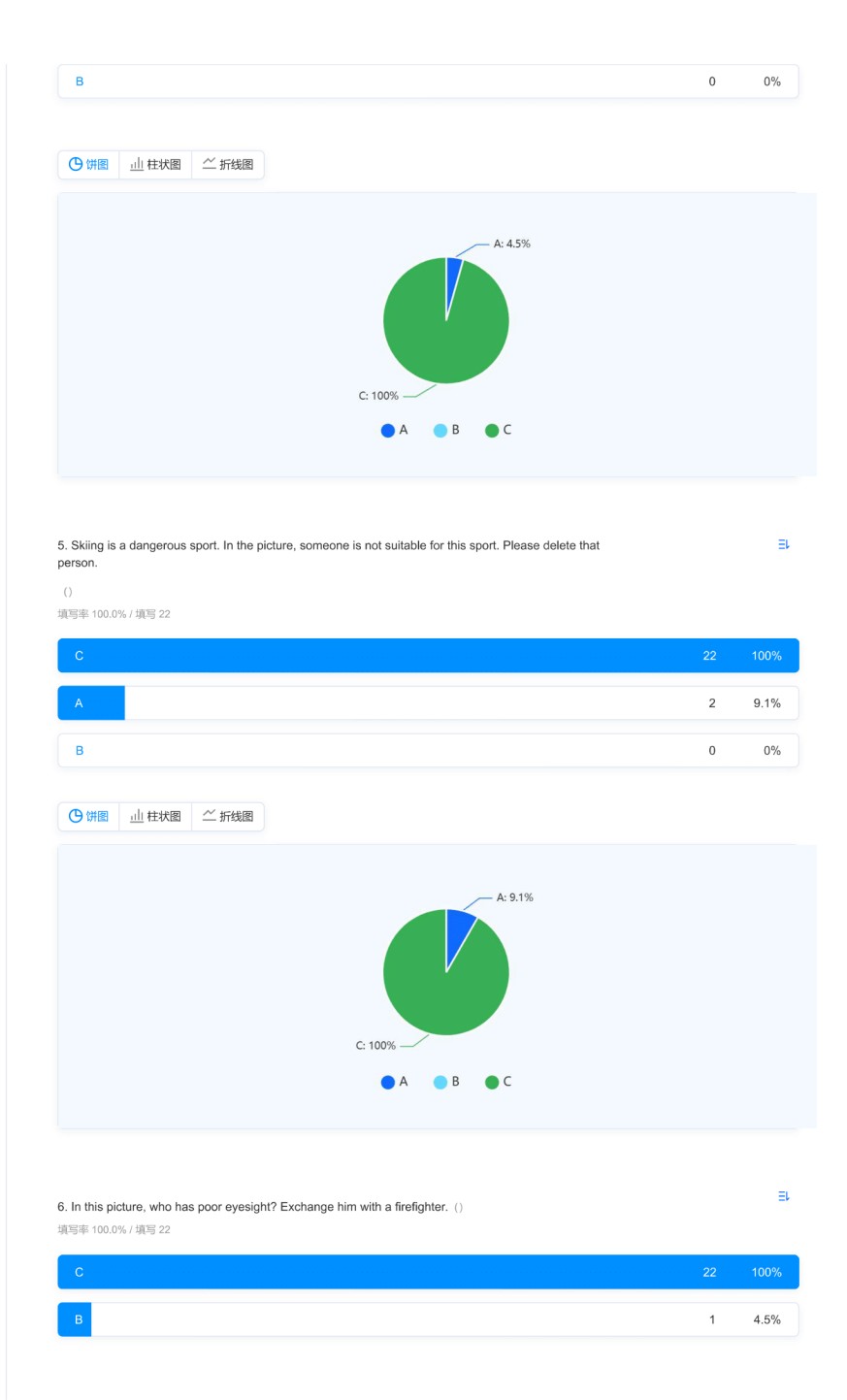

Figure A7: User Study Snapshot: Page 3

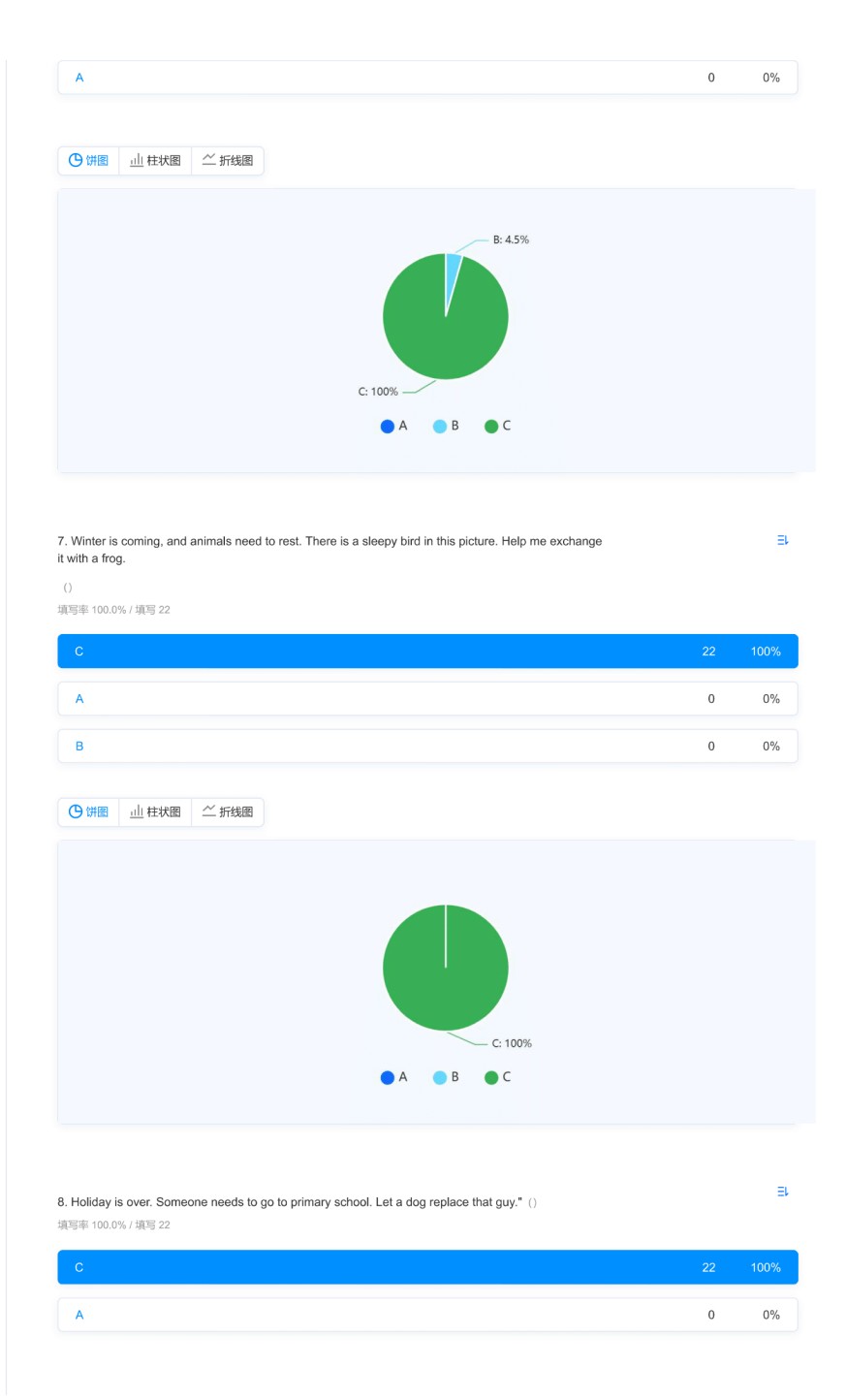

Figure A8: User Study Snapshot: Page 4

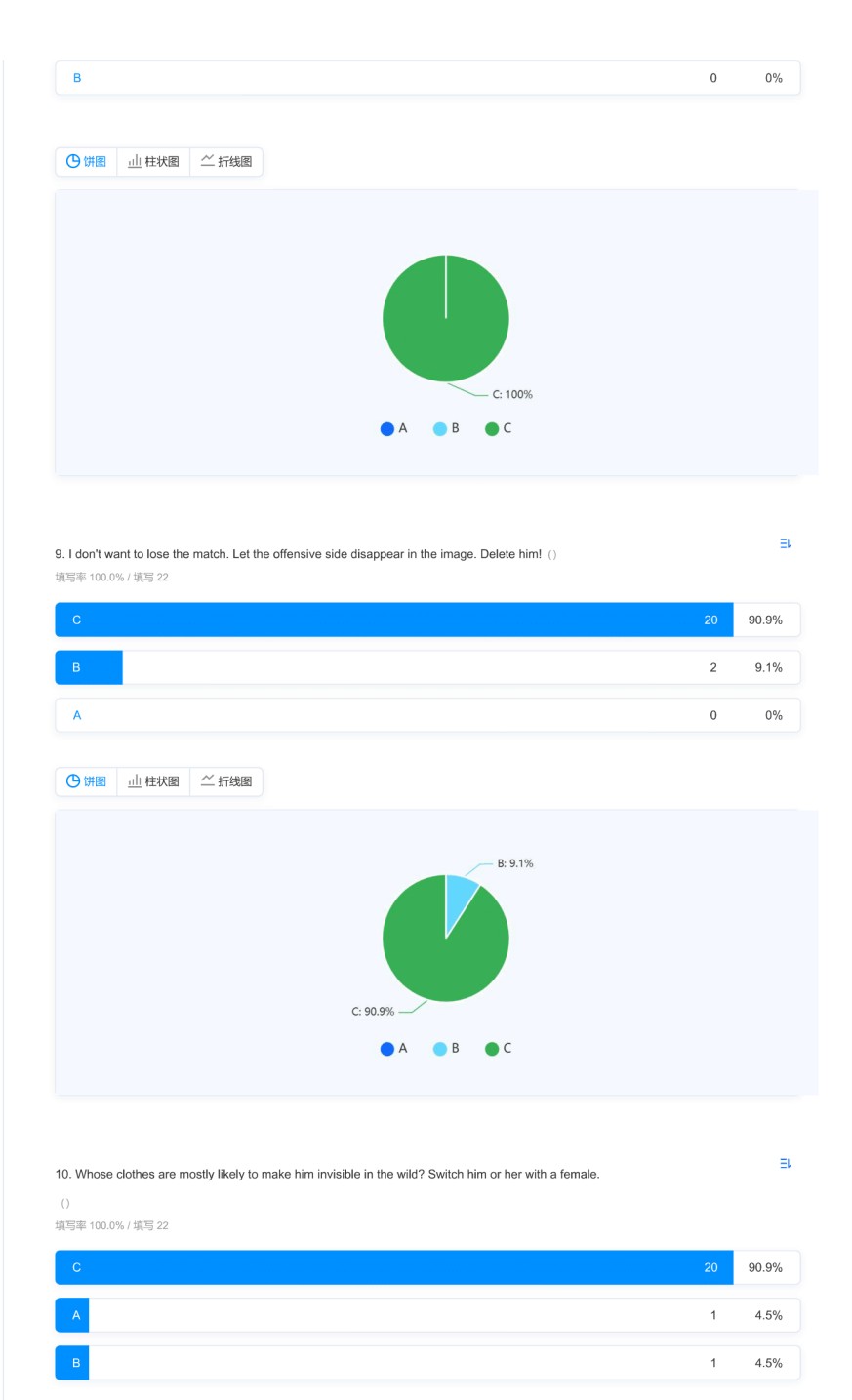

Figure A9: User Study Snapshot: Page 5

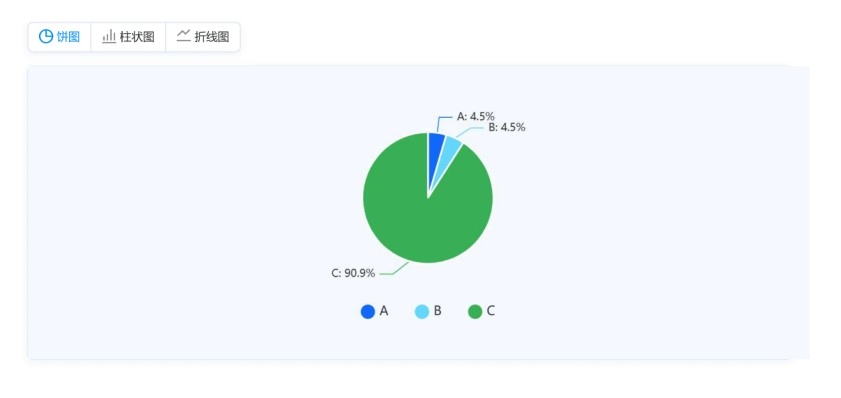

Figure A10: User Study Snapshot: Page 6

Tsu-Jui Fu, Wenze Hu, Xianzhi Du, William Yang Wang, Yinfei Yang, and Zhe Gan. Guiding instruction-based image editing via multimodal large language models. *arXiv preprint arXiv:2309.17102*, 2023. 5

Yuying Ge, Sijie Zhao, Jinguo Zhu, Yixiao Ge, Kun Yi, Lin Song, Chen Li, Xiaohan Ding, and Ying Shan. Seed-x: Multimodal models with unified multi-granularity comprehension and generation. *arXiv preprint arXiv:2404.14396*, 2024. 4

Qingdong He, Xueqin Chen, Chaoyi Wang, Yanjie Pan, Xiaobin Hu, Zhenye Gan, Yabiao Wang, Chengjie Wang, Xiangtai Li, and Jiangning Zhang. Reasoning to edit: Hypothetical instruction-based image editing with visual reasoning. *arXiv*, 2025. 1

Yuzhou Huang, Liangbin Xie, Xintao Wang, Ziyang Yuan, Xiaodong Cun, Yixiao Ge, Jiantao Zhou, Chao Dong, Rui Huang, Ruimao Zhang, et al. Smartedit: Exploring complex instruction-based image editing with multimodal large language models. In *Proceedings of the IEEE/CVF Conference on Computer Vision and Pattern Recognition*, pp. 8362–8371, 2024. 1, 5

Ying Jin, Pengyang Ling, Xiaoyi Dong, Pan Zhang, Jiaqi Wang, and Dahua Lin. Reasonpix2pix: instruction reasoning dataset for advanced image editing. In *CVPRW*, 2024. 1

Yuanzhi Li, Sébastien Bubeck, Ronen Eldan, Allie Del Giorno, Suriya Gunasekar, and Yin Tat Lee. Textbooks are all you need ii: phi-1.5 technical report. *arXiv preprint arXiv:2309.05463*, 2023. 5

Xinlong Wang, Xiaosong Zhang, Zhengxiong Luo, Quan Sun, Yufeng Cui, Jinsheng Wang, Fan Zhang, Yueze Wang, Zhen Li, Qiying Yu, et al. Emu3: Next-token prediction is all you need. *arXiv preprint arXiv:2409.18869*, 2024. 4

Yecheng Wu, Zhuoyang Zhang, Junyu Chen, Haotian Tang, Dacheng Li, Yunhao Fang, Ligeng Zhu, Enze Xie, Hongxu Yin, Li Yi, et al. Vila-u: a unified foundation model integrating visual understanding and generation. *arXiv preprint arXiv:2409.04429*, 2024. 4

Shitao Xiao, Yueze Wang, Junjie Zhou, Huaying Yuan, Xingrun Xing, Ruiran Yan, Chaofan Li, Shuting Wang, Tiejun Huang, and Zheng Liu. Omnigen: Unified image generation, 2024. URL https://arxiv.org/abs/2409.11340. 4, 5

Ling Yang, Bohan Zeng, Jiaming Liu, Hong Li, Minghao Xu, Wentao Zhang, and Shuicheng Yan. Editworld: Simulating world dynamics for instruction-following image editing. *arXiv preprint arXiv:2405.14785*, 2024. 1

Kai Zhang, Lingbo Mo, Wenhu Chen, Huan Sun, and Yu Su. Magicbrush: A manually annotated dataset for instruction-guided image editing. *Advances in Neural Information Processing Systems*, 36:31428–31449, 2023. 1

Xiangyu Zhao, Peiyuan Zhang, Kexian Tang, Hao Li, Zicheng Zhang, Guangtao Zhai, Junchi Yan, Hua Yang, Xue Yang, and Haodong Duan. Envisioning beyond the pixels: Benchmarking reasoning-informed visual editing. *arXiv preprint arXiv:2504.02826*, 2025. 1