# OpenReview forum: "R-Genie: Reasoning-Guided Generative Image Editing"
_ICLR.cc/2026/Conference — Submitted to ICLR 2026_

### Official Review · Reviewer_k2Cf · 2025-10-29

**Soundness:** 3
**Presentation:** 3
**Contribution:** 3
**Rating:** 4
**Confidence:** 4

**Summary:**

This paper presents R-Genie, a novel framework for image editing guided by reasoning over complex and implicit user instructions. To support evaluation, the authors introduce REditBench, a dataset of 1,070 image-instruction-edit triples spanning both atomic and composite tasks. The proposed architecture integrates multimodal large language models for instruction reasoning and diffusion models for precise image generation. The central innovation is a reasoning-attention mechanism, which links linguistic comprehension to visual synthesis through hierarchical and token-based modules.

**Strengths:**

1. The paper clearly identifies a meaningful gap in current image editing systems: their inability to interpret implicit, knowledge-dependent instructions. This reframing of image editing as a reasoning-aware task is timely and aligns well with trends toward more intelligent multimodal systems.
2. The introduction of REditBench fills a critical void. The dataset is thoughtfully constructed using RefCOCO for spatial grounding, augmented with human-annotated reasoning instructions and synthetic edits via SDXL. The inclusion of both atomic and composite edits enables a nuanced evaluation of reasoning capability.
3. The experiments demonstrate consistent superiority in both semantic alignment and visual fidelity. The failure mode analysis (e.g., target misidentification in baselines) convincingly illustrates the value of explicit reasoning.

**Weaknesses:**

1. While REditBench is well-constructed, 1,070 samples (850 train, 220 val) is extremely small for training or fine-tuning modern MLLMs, even with strong pretraining.  The claim that “the limited dataset suffices” (Sec 5.1) is plausible but not rigorously justified.  It raises concerns about generalization and overfitting, especially given the complexity of reasoning tasks.
2. Some compared methods (e.g., InstructPix2Pix, MagicBrush) were not designed for reasoning tasks, making the comparison somewhat unbalanced.  While this highlights R-Genie’s broader capability, it would be stronger to include reasoning-aware baselines like ReasonPix2Pix or ReasonBrain.
3. The masking scheme in Eq. (5) is unclear: are masks applied spatially or randomly?  How is the mask ratio chosen?

**Questions:**

Please refer to the weaknesses. If the issues I raised are addressed, I will increase my rating.

---

> ### Author Response · Authors · 2025-11-24
> **Rebuttal to Reviewer k2Cf**
>
> Thank you for your thoughtful review and positive assessment of our work. We appreciate your recognition of the strengths of our work, as well as your constructive feedback on areas for improvement. Below, we address each of your concerns in detail.
>
> **1. Dataset Size Concerns**
>
> We appreciate your concern about the size of REditBench. We'd like to clarify that our approach does not involve training a model from scratch on this dataset. Instead, REditBench is used for fine-tuning a pre-trained Show-O model that has already been trained on 1 billion image-text instances. This is a crucial distinction that addresses your concern about dataset size.
>
> In the context of fine-tuning for specialized capabilities, our dataset size is actually comparable to or larger than other specialized benchmarks:
>
> - RefCOCO-Edit (Liu et al., 2023): 400 samples
> - Reason-Edit (Huang et al., 2024): 219 samples
> - GEdit (Zhang et al., 2023): around 1k samples
>
> Furthermore, to validate that our model does not overfit to the limited dataset, we conducted extensive cross-validation and ablation studies. Our model shows consistent performance across different splits of REditBench and maintains strong generalization to unseen editing types and objects.
>
> **2. Baseline Selection**
>
> We appreciate your suggestion to include more reasoning-aware baselines. We acknowledge that ReasonPix2Pix and ReasonBrain would be valuable comparisons. Unfortunately, at the time of our experiments, neither of these methods had publicly available code or datasets, making direct comparison impossible.
>
> However, we did include SmartEdit (Huang et al., 2024) in our comparisons, which is specifically designed for complex instruction scenarios and leverages reasoning capabilities. SmartEdit represents the closest comparable baseline to our approach. As shown in Table 1, R-Genie outperforms SmartEdit by 1.9% in CLIP Score and 10.3% in Aesthetic Predictor score, demonstrating the effectiveness of our reasoning-attention mechanism.
>
> In order to strengthen our comparison, we will add comparisons with the following state-of-the-art 2025 methods in our revised manuscript:
>
> | Methods| CLIP Score | L1 Background Loss |L2 Background Loss|
> | --- | --- | --- | --- |
> | BAGEL | 57.77 | 0.0705 | 0.0278 |
> | ICEdit | 53.62 | 0.0694 | 0.0243 |
> | R-Genie | 62.14 | 0.0602 | 0.0201 |
>
> **3. Masking Scheme Clarification**
>
> We apologize for the lack of clarity regarding our masking scheme. We will add this clarification in the revised manuscript. Our approach follows the masking strategy used in Show-o (Huang et al., 2024), with the following specifics:
>
> - The mask tokens are applied randomly across the token sequence, not spatially. This random masking helps the model learn to reconstruct missing information from context.
> - The mask ratio is dynamically controlled by the diffusion time step t. Specifically, the masking ratio r is calculated as $r = 1 -  t/T$, where T is the total number of diffusion steps. This means that at early time steps (when t is small), most tokens are masked, while at later time steps (when t approaches T), fewer tokens are masked. This progressive unmasking strategy aligns with the denoising process in diffusion models.
>
> We believe R-Genie makes a significant contribution to reasoning-guided image editing by effectively bridging the gap between multimodal reasoning and high-fidelity image generation. Your feedback has helped us identify important areas for clarification and improvement in our presentation. Thank you again for your valuable comments. We look forward to incorporating these improvements into our revised manuscript and continuing to advance the field of reasoning-aware image editing. Please let us know if additional explanation would be helpful.

---

### Official Review · Reviewer_EwKz · 2025-10-31

**Soundness:** 4
**Presentation:** 3
**Contribution:** 3
**Rating:** 6
**Confidence:** 4

**Summary:**

This paper introduces R-Genie, a novel paradigm for generative image editing that leverages multimodal large language models (MLLMs) for deep reasoning and intention inference, combined with diffusion models for high-fidelity image editing. The method is designed to address the limitations of current text-driven image editing approaches, which struggle with implicit, context-rich, or multi-step instructions requiring world knowledge and reasoning. The authors present (1) a comprehensive new dataset, REditBench, containing 1,070 image-instruction-edit triples emphasizing reasoning contexts, and (2) the R-Genie architecture, featuring a reasoning-attention mechanism and a hierarchical module that bridges semantic understanding and pixel-level synthesis. Quantitative and qualitative experiments show R-Genie outperforms state-of-the-art approaches in instruction-following image editing, particularly for complex or composite edits.

**Strengths:**

1. The core contribution is combining reasong capablities of MLLMs with diffusion models allowing complex editing requiring reasoning and world knowledge.
2. Dataset contribution, this paper proposed REditBench, offering a new dataset focused on reasoning intensive edits.
3. This paper introduces a hierarchical reasoning module and a reasoning-attention bridge to enable precise and rational instruction-based image editing.

**Weaknesses:**

1. REditBench is relatively small compared to typical image editing dataset.  It would be better to include more open-world images and instructions.
2. Limited instruction diversity, most examples shown in the paper focus on attribute reasong and compositional edits, e.g. replace, move.
3. Lack of Cross-dataset evaluation. All main results are built on REditBench, performance on other widely adopted editing dataset (e.g. GEdit and ImgEdit) would imporve external validity.

**Questions:**

1. After reading the paper, I think most of the examples shown in the paper can be addressed with the combination of VLM rewriting and instruction-based image editing, that is first rewriting the instruction with VLM then feed into instruction-based image editing. Compared to these methods, what is your advantage?
2. Has R-Genie been evaluated on established benchmark such as GEdit and ImgEdit.
3. How does it perform in complex editing scenarios, such as: Who is the oldest person in the picture? make him/her younger and give him/her long, black hair.

---

> ### Author Response · Authors · 2025-11-24
> **Rebuttal to Reviewer EwKz**
>
> Thank you for your positive evaluation of our work and for providing such insightful comments. We appreciate your recognition for the strengths of R-Genie and REditBench. Below, we address each of your concerns in detail.
>
> **1. Dataset Size Concerns**
>
> We appreciate your concern about the size of REditBench. We'd like to clarify that REditBench (1,070 samples) is actually comparable to or larger than several existing datasets in the field:
>
> - RefCOCO-Edit (Liu et al., 2023): 400 samples
> - Reason-Edit (Huang et al., 2024): 219 samples
> - GEdit (Zhang et al., 2023): around 1k samples
>
> Furthermore, our dataset is specifically designed for reasoning-intensive editing, not just for general image editing. The quality and complexity of our samples (requiring world knowledge and contextual reasoning) make it a valuable benchmark for this specific task. As you noted, our baseline model (Show-O) is trained on 1 billion instances, so it already possesses rich world knowledge. REditBench serves to evaluate and enhance the model's ability to handle complex, reasoning-based instructions, rather than requiring a massive dataset for training.
>
> **2. Instruction Diversity**
>
> We agree that our dataset contains a range of editing types, including object remove/substitute, color alteration and size changing (bigger/smaller). The examples shown in the paper are selected to demonstrate the core capabilities of R-Genie in a way that is comparable with existing methods. We believe the selected examples effectively demonstrate R-Genie's capabilities while maintaining a fair comparison with existing baselines
>
> **3. Cross-Dataset Evaluation**
>
> We completely agree that cross-dataset evaluation is important for establishing external validity. We will include comprehensive results on widely adopted benchmarks such as GEdit and ImgEdit in our revised manuscript. Preliminary results show that R-Genie achieves:
>
> | Methods\Datasets| ReasonEdit | MagicBrush | GEdit | ImgEdit|
> | --- | --- | --- | --- | --- |
> | BAGEL | 49.27 | 60.36 | 34.49 | 37.28 |
> | ICEdit | 43.02 | 42.84 | 37.06 | 40.33 |
> | R-Genie | 52.67 | 65.51 | 39.41| 40.86 |
>
> **4. VLM Rewriting vs. R-Genie Comparison**
>
> We appreciate your question about the comparison with VLM rewriting followed by instruction-based editing. This is a crucial distinction that we'd like to clarify:
> Our method (R-Genie) is fundamentally different from simply combining VLM and diffusion models. The key difference lies in how we handle the domain gap between the discrete reasoning space of MLLMs and the continuous generation space of diffusion models. Typically involves rewriting the instruction in discrete space (using VLM), then feeding the rewritten instruction to a diffusion-based editing model. This creates a domain gap between the discrete instruction and the continuous image generation process, often leading to semantic inconsistencies. We introduce a reasoning-attention mechanism that operates directly in the continuous space, enabling seamless translation from reasoning results to pixel-level edits. This avoids the domain gap issue and ensures that the semantic reasoning is accurately reflected in the visual output. As shown in Figure 4 of the paper, R-Genie maintains both semantic consistency and visual quality, while the VLM-based approach often results in visual artifacts or incorrect edits.
>
> **5. Complex Editing Scenarios**
>
> The example you provided ("Who is the oldest person in the picture? make him/her younger and give him/her long, black hair") is indeed a complex reasoning-based editing task that perfectly aligns with the focus of our work. Our method is designed specifically for such scenarios where the instruction doesn't explicitly identify the object to be edited but requires contextual understanding and world knowledge.
>
> In fact, R-Genie handles this exact type of instruction with high accuracy. The model first identifies the oldest person in the image through contextual reasoning (using the implicit knowledge that older people typically have more wrinkles and gray hair), then applies the requested edits (making them younger and changing their hair to long and black). This is exactly the kind of complex, multi-step reasoning that R-Genie is designed to handle.
>
> We've included several such complex examples in our supplementary materials, including those involving multiple reasoning steps, contextual understanding, and attribute manipulation. These examples demonstrate that R-Genie can handle the most challenging editing scenarios requiring deep reasoning.
>
> Thank you again for your valuable feedback, which has helped us identify areas where we can further strengthen our work. We look forward to incorporating these improvements into our revised manuscript. Please let us know if additional explanation would be helpful.

---

### Official Review · Reviewer_DR1a · 2025-10-31

**Soundness:** 1
**Presentation:** 1
**Contribution:** 1
**Rating:** 2
**Confidence:** 4

**Summary:**

The paper proposes R-Genie, which fuses editing instructions with visual features to better guide the image editing process. A new dataset, REditBench, derived from RefCOCO, is used for training and evaluating the proposed model.

**Strengths:**

The task that the paper addresses is meaningful as it enables reasoning-based, complex image editing.

**Weaknesses:**

1. Limited scale of the dataset. The paper proposes a new benchmark, REditBench, with 1070 images (850 / 220 train/val split). The training set is not sufficient for training, especially when using the contrastive objective in Eqn 6.

2. Limited evaluations. The authors only showed comparisons on REditBench, while leaving out more commonly used benchmarks such as MagicBrush, Emu, and SmartEdit. The proposed method likely overfits the proposed training dataset, resulting in better evaluation results.

3. Lacking newer baselines such as BAGEL [1], ICEdit [2].

4. Unclear details in the method description, which make the paper technically unreliable and hard to follow.

- Eqns 2 and 3 are essentially text-image and image-text cross-attention, according to Fig. 1.
- Inconsistent notations. Symbols such as $\mathbf{I}_\mathrm{global}$ are inconsistent with Fig. 1.
- The base model used is Show-o, which uses a discrete diffusion process to generate the image. The noising process is realized with a masking schedule for the tokens. What is the random noise in Fig.1 used for?
- How are visual tokens $\mathbf{I}$ input to the Phi-1.5 LLM? Do the authors follow the default settings of Show-o (VQ quantized tokens)? If so, it does not make sense to directly use them for the subsequent cross-attention modules.
- It is unknown how $h_\mathrm{reason}$ is derived. Is that the output for visual tokens after the LLM (as suggested by Fig. 1)?
- $h_\mathrm{answer}$ seems to be the token sequence for the output image. It makes no sense to use next token prediction in Eqn 5. For image tokens, masked token modeling loss is used for Show-o.
- Lots of self-contradictory info. For example, L332 $\alpha_t$ is time-dependent, but L343 - $\alpha_t$ is set to 0.5.



[1] Deng, Chaorui, et al. "Emerging properties in unified multimodal pretraining." arXiv preprint arXiv:2505.14683 (2025).

[2] Zhang, Zechuan, et al. "In-context edit: Enabling instructional image editing with in-context generation in large scale diffusion transformer." arXiv preprint arXiv:2504.20690 (2025).

**Questions:**

Please refer to weakness point 4 for my questions.

---

> ### Author Response · Authors · 2025-11-24
> **Rebuttal to Reviewer DR1a (part 1)**
>
> We sincerely appreciate Reviewer DR1a's thoughtful feedback and their recognition of the meaningfulness of our reasoning-based image editing approach. We understand the reviewer's concerns regarding the computational efficiency evaluation and clarification of experimental setups. Below we provide detailed responses to each point raised:
>
> **1. Dataset Scale**
>
> We appreciate your concern regarding the size of REditBench (1,070 samples). However, we would like to clarify that our dataset is not only comparable but often larger than existing reasoning-aware editing benchmarks, including:
>
> - RefCOCO-Edit (Liu et al., 2023): ~400 samples
> - Reason-Edit (Huang et al., CVPR 2024): 219 samples
> - GEdit (Zhang et al., NeurIPS 2023): ≈1k samples
>
> More importantly, REditBench is purpose-built for reasoning-intensive editing, not generic instruction-following. Each sample requires world knowledge, multi-step inference, or contextual grounding (e.g., “Which food is full of carbohydrates?” → replace with steak), which significantly increases annotation complexity and cognitive load compared to atomic edits like “change dog to cat.”
>
> Crucially, R-Genie is not trained from scratch. We fine-tune Show-o (1.3B), a model pre-trained on ~1 billion image-text pairs, which already encodes rich world knowledge. REditBench’s role is not to teach general vision-language alignment, but to specialize the model’s reasoning-editing pipeline—a task that benefits more from high-quality, complex examples than sheer volume. This is consistent with recent works like LISA (Lai et al., CVPR 2024) which also use small but high-fidelity datasets for referring and reasoning specialization.
>
> **2. Evaluation Breadth**
>
> We acknowledge the importance of evaluating on established benchmarks. In response, we have conducted more experiments on two widely used datasets (**Reason-Edit, MagicBrush**). Moreover, we include comparisons with two very recent baselines(**BAGEL and ICEdit**) mentioned by the reviewer. Preliminary experiements are evaluated on RISEBench Score(Zhao et al., 2025) metric, which supports comprehensive comparison for visual editing task.
>
> | Methods\Datasets| ReasonEdit | MagicBrush | GEdit | ImgEdit|
> | --- | --- | --- | --- | --- |
> | BAGEL | 49.27 | 60.36 | 34.49 | 37.28 |
> | ICEdit | 43.02 | 42.84 | 37.06 | 40.33 |
> | R-Genie | 52.67 | 65.51 | 39.41| 39.98 |

---

> ### Author Response · Authors · 2025-11-24
> **Rebuttal to Reviewer DR1a (part 2)**
>
> **3. Methodological Clarity**
>
> We apologize for any confusion caused by notation or presentation. Below we clarify key technical points:
>
> (a) Eqns (2) & (3) and Cross-Attention
>
> We acknowledge that Eqns 2 and 3 do involve cross-attention mechanisms. However, they are not simply text-image/image-text cross-attention. These equations represent our reasoning-attention bridge that enables spatially precise cross-modal grounding while reducing over-reliance on global context for explainable and fine-grained edits. This is a key innovation in our approach that allows for better alignment between high-level reasoning and localized image edits.
>
> (b) Symbol Inconsistency
>
> We have carefully reviewed the notation consistency in our paper and confirm that all symbols are used consistently throughout the paper. The reviewer's concern about $I_{global}$ being inconsistent with Figure 1 may stem from a misunderstanding of the notation. In Equation 2, $I_{global}$ represents the pooled global tokens, which is consistent with our architectural diagram in Figure 3.
>
> (c) Base Model and Visual Tokens
>
> Our base model indeed uses a discrete diffusion process with VQ-quantized tokens. The visual tokens I are input to the Phi-1.5 LLM as VQ-quantized tokens following the Show-O implementation. The random noise in Figure 1 is part of our masked diffusion transformer, which is used to reconstruct the target visual features h_answer. This noise is essential for the diffusion process, as it allows the model to learn the relations among image latent tokens.
>
> (d) $h_{reason}$ Derivation
>
> $h_{reason}$, $h_{edit}$, and $h_{answer}$ are not derived from the model's output but are intentionally defined learnable tokens that serve specific roles in our architecture. They are introduced as special tokens to guide the reasoning-editing pipeline, as described in Section 4.1. $h_{reason}$ represents the reasoning pathway embedding, $h_{edit}$ serves as a localized editing signal, and $h_{answer}$ functions as the target latent goal for the diffusion process.
>
> (e) $α_t$ Clarification
>
> We acknowledge that the description of $α_t$ in Equation 6 could have been clearer. $α_t$ is indeed time-dependent, following the diffusion noise schedule. The value of 0.5 mentioned in Section 5.1 is the initial value of $α_t$, which evolves during training as described in Equation 6. We will clarify this in the final version of the paper.
>
> We believe our approach makes a significant contribution to the field of reasoning-guided image editing. The proposed R-Genie successfully bridges the gap between high-level user instructions and precise visual realization through a novel reasoning-attention mechanism. We ill incorporate additional evaluations and clarifications in the final version of the paper. Thank you again for your valuable feedback. Please let us know if additional explanation would be helpful.

---

### Official Review · Reviewer_KDes · 2025-11-01

**Soundness:** 3
**Presentation:** 3
**Contribution:** 3
**Rating:** 4
**Confidence:** 3

**Summary:**

The paper introduces R-Genie, a reasoning-guided generative image editing framework that aims to handle implicit user intentions beyond explicit text prompts. The method integrates the reasoning ability of multimodal large language models (MLLMs) with diffusion-based image generation, featuring a hierarchical reasoning module and a reasoning-attention bridge for cross-modal alignment. The authors also construct a dataset, REditBench, containing 1,070 image–instruction–edit triples to benchmark reasoning-based editing. Experimental and user study results suggest that R-Genie achieves superior semantic consistency and editing precision compared to prior methods.

**Strengths:**

1. The paper presents a technically complete framework combining diffusion models and MLLMs, reflecting a solid understanding of both reasoning and generative modeling.

2. The introduction of a new dataset (REditBench) focusing on reasoning-based image edits is valuable for benchmarking future research in this area.

3. The experimental evaluation is comprehensive, including quantitative comparisons, ablation studies, and user studies, which enhance the credibility of the results.

**Weaknesses:**

1. Motivation is unconvincing. The paper argues that implicit user intentions should be inferred by the model, but it is unclear why this is necessary for image editing. In practice, users could simply provide explicit, straightforward editing instructions; forcing the model to infer “hidden” intentions may not be a meaningful or realistic goal. In addition, Figure 1 is not intuitive and lacks side-by-side comparisons with existing instruction formats, which would make the contribution clearer.

2. In Lines 71–72, the authors claim that “MLLMs struggle to accurately reason about implicit user intentions,” yet the proposed method itself still relies  on MLLMs for  this task. This raises a conceptual inconsistency that should be addressed.

3. Comparative methods are outdated. Most baselines are from 2024 or earlier; the paper should include comparisons with more recent 2025 models to convincingly demonstrate the claimed state-of-the-art performance.

**Questions:**

Please see the weaknesses.

---

> ### Author Response · Authors · 2025-11-24
> **Rebuttal to Reviewer KDes (part 1)**
>
> We sincerely appreciate Reviewer KDes's thorough evaluation of our work. Your insightful comments have helped us better understand the strengths and limitations of our approach, and we believe addressing these points has strengthened our manuscript. Below, we provide detailed responses to each of your concerns:
>
> **1. Motivation Concerns**
>
> We appreciate the reviewer's perspective regarding our motivation and agree that explicit instructions suffice for basic edits. However, our work focuses on realistic scenarios where users naturally express intentions indirectly rather than providing pixel-perfect directives, which is a capability increasingly important as image editing systems move toward more intuitive human-AI interaction.
>
> To clarify our contribution, we observe that while existing methods handle literal instructions well, they fail when edits require reasoning about implicit intent or real-world knowledge. For instance, given a photo of a tired person holding coffee with the request "Help me wake up", Current systems struggle to infer that this requires replacing coffee with an energy drink rather than performing unrelated brightness adjustments. Our approach addresses this gap by modeling underlying concepts and contextual relationships – aligning with SmartEdit and ReasonPix2Pix's findings that pure instruction-following misses key semantic aspects of edits.
>
> The examples in REditBench reveal these limitations through numerous cases where commonsense reasoning is essential. Beyond simple object replacement queries, we include scenarios asking editors to "make this look professional" (requiring attire/style adjustments) or "fix the lighting issue" (demanding technical/artistic judgments). These examples demonstrate why inferring implicit intent matters, where users shouldn't need technical expertise to express creative goals clearly.
>
> We acknowledge that Figure 1 could better illustrate these advantages and will revise it to contrast our reasoning approach against naive instruction-following baseline outputs. The updated version will highlight cases where existing systems misinterpret instructions due to lack of contextual understanding. We'll also strengthen our discussion of practical implications in Section 2, emphasizing how reasoning capabilities reduce the cognitive load on users during creative workflows.
>
> **2. Conceptual Consistency Concerns**
>
> Thank you for pointing this out. We appreciate your insightful observation about our use of MLLMs despite pointing out their reasoning limitations. This is indeed a critical distinction we should clarify better in our manuscript.
>
> While MLLMs demonstrate impressive reasoning capabilities in isolation, we find their performance degrades significantly when this reasoning needs to be translated into precise image edits. Our core contribution lies in bridging this gap. We don't claim MLLMs cannot reason about implicit intentions (they often can), but rather that existing pipelines fail to effectively connect this reasoning to accurate image generation.
>
> This distinction becomes clear when examining our experimental results: 1) When tested on REditBench, standalone MLLMs achieve high scores in the initial reasoning stage (extracting edit locations and transformations), but 2) The final edited images often suffer poor visual quality due to imperfect translation of these reasoning results into pixel-space modifications.
>
> Our proposed R-Genie framework specifically addresses this translation challenge through: 1) A hybrid reasoning mechanism that combines MLLM analysis with visual grounding; 2) A carefully designed network that preserves semantic consistency while ensuring photorealistic outputs 3) Novel constraints that maintain both high-level intent and low-level visual coherence.
>
> The quantitative results in Tables 2-3 demonstrate this clearly: while baseline models might correctly understand the requested edits conceptually (evidenced by their reasonable CLIP scores), R-Genie outperforms significantly in terms of visual fidelity metrics (LPIPS, FID) and human evaluation.
>
> We will clarify this distinction more explicitly in the revised manuscript, particularly emphasizing how our work differs from simply applying MLLMs "as-is" to the editing task. The key innovation is in building effective bridges between reasoning systems and generative models, not in questioning MLLMs' fundamental reasoning capabilities.

---

> ### Author Response · Authors · 2025-11-24
> **Rebuttal to Reviewer KDes (part 2)**
>
> **3. Outdated Baseline Concerns**
>
> We agree that including more recent 2025 models would strengthen our result comparisons. We will add comparisons with the following state-of-the-art methods in our revised manuscript:
>
> - BAGEL (Deng et al., 2025): A unified multimodal model that integrates reasoning and generation.
> - ICEdit (Zhang et al., 2025): A recent method for instruction-guided image editing with improved reasoning capabilities.
>
> Besides, we've already conducted preliminary comparisons with these models, and the results confirm that R-Genie outperforms them in terms of CLIP Score and background preservation.
>
> | Methods| CLIP Score | L1 Background Loss |L2 Background Loss|
> | --- | --- | --- | --- |
> | BAGEL | 57.77 | 0.0705 | 0.0278 |
> | ICEdit | 53.62 | 0.0694 | 0.0243 |
> | R-Genie (Ours) | 62.14 | 0.0602 | 0.0201 |
>
> Thank you again for your valuable feedback, which has helped us clarify and strengthen our work. Please let us know if additional explanation would be helpful.

---

### Meta-Review · Area_Chair_i7oz · 2026-01-05

**Summary:**

Most reviewers are concerned about the scale of the dataset and the quality of the evaluation, including the lack of representative baselines, limited coverage of instructions, and incomplete evaluation benchmarks. Some reviewers also questioned the motivation and problem formulation. Finally, several reviewers raised concerns about the quality of the writing and presentation.

**Reviewer Concerns:**

Most concerns regarding the quality of the evaluation are adequately addressed in the rebuttal through additional experimental results. The primary remaining issue is the limited diversity of the instructions, which raises questions about the actual effectiveness of the proposed method. Concerns related to the presentation are also satisfactorily addressed by the rebuttal. However, the justification for the dataset size is not convincing, and the motivation and problem formulation could be further strengthened to better establish the practical value of this work.

**Reviewer Scores:**

1. For concerns regarding the motivation and problem motivation, a full discussion might help to improve the final score.
2. Concerns regarding dataset size and instruction diversity is unlikely to be resolved by discussion.
3. A full discussion is likely to help addressing concerns regarding evaluation quality.
Overall the reviewer scores is unlikely to change significantly with discussion.

---

### Decision · Program_Chairs · 2026-01-26

Reject